# *Mtfp1* ablation enhances mitochondrial respiration and protects against hepatic steatosis

Cecilia Patitucci[1,8], Juan Diego Hernández-Camacho[1,8], Elodie Vimont[1], Sonny Yde [1], Thomas Cokelaer [2,3], Thibault Chaze[4], Quentin Giai Gianetto[3,4], Mariette Matondo [3], Anastasia Gazi [5], Ivan Nemazanyy [6], David A. Stroud [7], Daniella H. Hock[7], Erminia Donnarumma[1] & Timothy Wai [1] ✉

Hepatic steatosis is the result of imbalanced nutrient delivery and metabolism in the liver and is the first hallmark of Metabolic dysfunction-associated steatotic liver disease (MASLD). MASLD is the most common chronic liver disease and involves the accumulation of excess lipids in hepatocytes, inflammation, and cancer. Mitochondria play central roles in liver metabolism yet the specific mitochondrial functions causally linked to MASLD remain unclear. Here, we identify Mitochondrial Fission Process 1 protein (MTFP1) as a key regulator of mitochondrial and metabolic activity in the liver. Deletion of *Mtfp1* in hepatocytes is physiologically benign in mice yet leads to the upregulation of oxidative phosphorylation (OXPHOS) activity and mitochondrial respiration, independently of mitochondrial biogenesis. Consequently, liver-specific knockout mice are protected against high fat diet-induced steatosis and metabolic dysregulation. Additionally, *Mtfp1* deletion inhibits mitochondrial permeability transition pore opening in hepatocytes, conferring protection against apoptotic liver damage in vivo and ex vivo. Our work uncovers additional functions of MTFP1 in the liver, positioning this gene as an unexpected regulator of OXPHOS and a therapeutic candidate for MASLD.

Metabolic dysfunction-associated fatty liver disease (MASLD) formally known as NAFLD (non-alcoholic fatty liver disease)[1], is the most common chronic liver disease in industrialized countries whose incidence is rapidly expanding worldwide[2]. MASLD is a frequent comorbidity of type 2 diabetes and obesity with its prevalence calculated at ~30% in the general population and 80% among obese people. MASLD encompasses a spectrum of pathologies ranging from simple steatosis characterized by triglyceride accumulation in hepatocytes, to metabolic dysfunction-associated Steatohepatitis (MASH), whose hallmarks are inflammation and fibrogenesis, which can further progress into cirrhosis and hepato-cellular carcinoma (HCC), the deadliest form of liver cancer[3].

Considerable efforts have been made in recent decades to better understand the mechanisms of MASLD progression and therapeutic

[1]Institut Pasteur, Mitochondrial Biology Group, CNRS UMR 3691, Université Paris Cité, Paris, France. [2]Institut Pasteur, Biomics Technological Platform, Université Paris Cité, Paris, France. [3]Institut Pasteur, Bioinformatics and Biostatistics Hub, Université Paris Cité, Paris, France. [4]Institut Pasteur, Proteomics Core Facility, MSBio UtechS, UAR CNRS 2024, Université Paris Cité, Paris, France. [5]Institut Pasteur Ultrastructural Bio Imaging, UTechS, Université Paris Cité, Paris, France. [6]Platform for Metabolic Analyses, SFR Necker, INSERM US24/CNRS UAR 3633, Paris, France. [7]Department of Biochemistry and Pharmacology, Bio21 Molecular Science and Biotechnology Institute, University of Melbourne, Victorian Clinical Genetics Services and Murdoch Children's Research Institute, Royal Children's Hospital, Melbourne, VIC, Australia. [8]These authors contributed equally: Cecilia Patitucci, Juan Diego Hernández-Camacho. ✉e-mail: timothy.wai@pasteur.fr

targets that might subsequently alleviate the burden of this spectrum of pathologies. The progression of MASLD is currently explained by a "multiple parallel-hit" hypothesis, which implicates the synergistic and concerted action of multiple events originating from various liver cell types[4]. In hepatocytes, oxidative stress and mitochondrial dysfunction have been suggested to contribute to hepatocyte damage and death, tissue inflammation and fibrosis[5]. This model highlights the complexity and heterogeneity of MASLD progression and underscores the central involvement of hepatocytes mitochondria in the progression of MASLD. Mitochondria are essential organelles that are deeply integrated in cellular homeostasis. Most famous for their production of ATP via oxidative phosphorylation (OXPHOS), mitochondria also regulate programmed cell death and inflammation through the sequestration and release of pro-apoptotic factors and pro-inflammatory molecules[6,7]. Yet, which of the multiple functions of mitochondria are directly implicated in the onset of steatosis, inflammation, hepatocyte death, and subsequent tissue remodeling has not been defined[8].

The relevance of mitochondria to liver function is highlighted by mitochondrial dysfunction observed in inborn and acquired liver pathologies: mutations in mitochondrial genes that cause genetic diseases manifest with liver dysfunction, while strong associations also exist between mitochondrial dysfunction and acquired liver diseases such as MASLD, viral hepatitis, and ischemic liver injury[9,10]. Mitochondrial respiration has been reported to decline during the progression of liver dysfunction in humans[11] and perturbation of mitochondrial structure has been reported in liver biopsies from patients with MASH[12], lending support to the notion that the maintenance of mitochondrial integrity is paramount to liver function. Promoting enhanced oxygen consumption with targeted uncouplers in the liver reduces the deleterious accumulation of hepatic lipid storage and steatosis in rodents and primates[13,14], yet broader system-wide mitochondrial uncoupling has catastrophic effects on other organs and is incompatible with life[15]. We recently showed that mitochondrial uncoupling in cardiac mitochondria is regulated by Mitochondrial Fission Process 1 (MTFP1), uncovering an additional role of this protein in bioenergetic regulation[16]. MTFP1 is a protein localized at the inner membrane (IMM) whose namesake derives from a putative role in mitochondrial fission in vitro[17,18] and has garnered interest as marker of liver dysfunction in humans. Retrospective studies revealed a link between *MTFP1* expression in tumoral tissue and patient survival in HCC[19] whose development is heavily influenced by MASLD progression[3]. Indeed, compelling evidence implicates MTFP1 in metabolic sensing and programmed cell death regulation in vitro[18,20,21], although its pertinence in the liver function and metabolism has never been explored.

Here, we report the generation of a liver-specific *Mtfp1* knockout mouse model (LMKO) and discover that, contrary to what we observed for the heart[16], MTFP1 is dispensable for organ function. We report that deletion of *Mtfp1* in vivo in hepatocytes enhances hepatic OXPHOS activity and confers protection against diet-induced liver steatosis, weight gain and systemic glucose dysregulation when mice are fed a high fat diet (HFD). In sum, our data reveal liver-specific effects of MTFP1 ablation in vivo that position this gene as a therapeutic candidate for MASLD.

## Results

### Generation and characterization of liver specific *Mtfp1* KO mouse model

To investigate the link between mitochondrial function and liver metabolism we generated a liver-specific KO mouse model in which we specifically deleted *Mtfp1* in post-natal hepatocytes (LMKO mice; Figs. 1A, S1A, B). Conditional mice (*Mtfp1^LoxP/LoxP*) previously generated on a C57Bl6/N background[16] were crossed to Alb-Cre recombinase transgenic mice (*Alb-Cre^{tg/+}*) to generate LMKO mice (*Alb-Cre^{tg/+}Mtfp1^{LoxP/LoxP}*). Genetic deletion of *Mtfp1* in LMKO mice was specific to the liver

(Fig. 1B) and caused a profound depletion of mRNA (Fig. 1C, Supplementary Dataset 1) and protein (Fig. 1D) expression in liver extracts. MTFP1 ablation in hepatocytes did not affect perinatal survival (Fig. 1E) and histological liver analysis performed on LMKO mice were unremarkable and indistinguishable from those of control littermates (Fig. 1F). Contrary to the ablation of MTFP1 in the heart[16], LMKO mice did not manifest any overt defects during their lifetime under normal chow diet (NCD). Comprehensive assessment of liver structure and function revealed no defects: liver mass (Fig. 1G) was unchanged in LMKO mice and the levels of circulating biomarkers of liver damage such as alanine aminotransaminase (ALAT) and aspartate aminotransaminase (ASAT) were not increased relative to those of control littermates (Fig. 1H). Circulating levels of cholesterol and triglycerides were normal (Fig. 1I) and metabolic cage performance was not impaired (Fig. 1J, S1C). Similarly, we observed no evidence of pathological gene expression changes by RNAseq analysis (Fig. S1D, Supplementary Dataset 1). In fact, control and LMKO livers revealed virtually no gene expression dysregulation (only 27 out of >25,000 transcripts) including no upregulation of established markers[22] of MASLD or HCC defined in humans (Fig. S1D, Supplementary Dataset 1). Taken together, our observations demonstrate that the deletion of *Mtfp1* in hepatocytes does not negatively impact the liver under basal conditions.

### *Mtfp1* deletion increases OXPHOS activity and mitochondrial respiration

Given our previous observations that MTFP1 ablation reduces bioenergetic efficiency of cardiac mitochondria[16], we decided to directly assess the impact of *Mtfp1* deletion on hepatic mitochondrial bioenergetics. To this end, we performed high resolution fluorrespirometry on isolated hepatic mitochondria isolated from control and LMKO mice, simultaneously measuring both oxygen consumption rates ($JO_2$) and mitochondrial membrane potential ($\Delta\Psi$) changes with Rhodamine 123 (RH-123)[23]. $JO_2$ and RH-123 were recorded from mitochondria incubated with respiratory substrates promoting the delivery of electrons to complex I (state 2; pyruvate, glutamate, and malate (PGM) or complex II (state 2; succinate and rotenone), or in the presence of palmitoyl-carnitine plus malate (state 2) in both the phosphorylating (state 3: ADP) and non-phosphorylating (state 4: oligomycin (OLGM) to inhibit ATP synthase) states (Fig. 2A). Notably, and in contrast to MTFP1-deficient cardiac mitochondria[16], respiration in LMKO liver mitochondria was significantly increased in phosphorylating (state 3) conditions in the presence of any of the respiratory substrates we tested. Pyruvate, glutamate, and malate led to a 49% increase in state 3 respiration and succinate and rotenone led to a 57% increase in respiration (Fig. 2B). Interestingly, we observed a 200% increase in state 3 respiration in the presence of palmitoyl carnitine, a fatty acid ester derivative, pointing to an increased efficiency of fatty-acid derived energy metabolism caused by liver-specific deletion of *Mtfp1*.

Additionally, LMKO liver mitochondria showed a higher respiratory control ratio (RCR) in the presence palmitoyl carnitine plus malate (Fig. 2C). Despite a marked increase in state 3 respiration, we did not observe a genotype-specific difference in mitochondrial membrane potential (Fig. 2D), which initially surprised us since increased oxygen consumption rates are typically accompanied by reduction in membrane potential due to dissipation of the protonmotive force via complex V (to synthesize ATP). The most parsimonious explanation for this result is that MTFP1 ablation promotes a commensurate increase in the activities of both cytochrome *c* oxidase (complex IV) and the ATP synthase (complex V). Indeed, when we measured the specific activities of complex IV (Fig. 2E) and complex V (Fig. 2F) in separate assays, we found a ~20% increase in LMKO mitochondria relative to control littermate controls, suggesting a similar contribution of both complexes to increase respiration while maintaining mitochondrial membrane potential. We further confirmed our findings

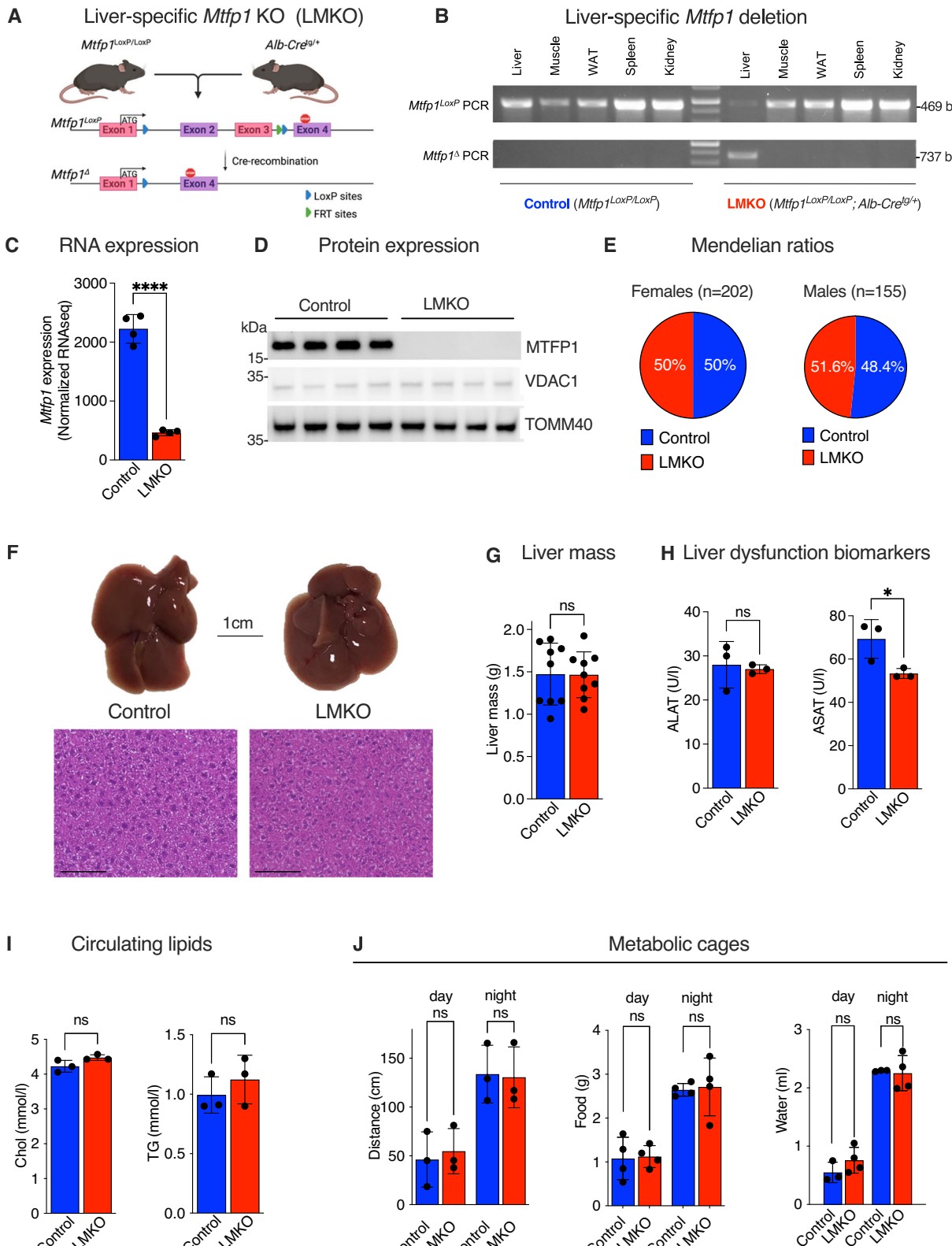

**A** Liver-specific *Mtfp1* KO (LMKO)

**B** Liver-specific *Mtfp1* deletion

**C** RNA expression

**D** Protein expression

**E** Mendelian ratios

Females (n=202)   Males (n=155)

**F**

Control   LMKO

**G** Liver mass

**H** Liver dysfunction biomarkers

**I** Circulating lipids

**J** Metabolic cages

by measuring oxygen consumption rates in mitoplasts supplied with either NADH, Cyt *c*, succinate, and rotenone to drive electron transport via Complex II or NADH, Cyt *c*, succinate and malonate to drive electron transport via Complex I (Fig. 2G). In both assays, oxygen consumption was elevated in LMKO liver mitochondria demonstrating that Complex IV activity is intrinsically augmented upon MTFP1 ablation independently of the protonmotive force. Together, these

observations implied either that MTFP1 ablation triggers a specific upregulation of components of the OXPHOS machinery or the induction of a general mitochondrial biogenesis response. To differentiate between these possibilities, we assessed mitochondrial mass using multiple molecular and imaging-based methods. Quantification of mitochondrial mass in primary hepatocytes isolated from control or LMKO mice expressing a genetically encoded matrix targeted YFP

**Fig. 1 | Hepatic deletion of *Mtfp1* in mice does not impair basal liver function.**
**A** Generation of liver-specific *Mtfp1* knockout mice (LMKO). Conditional *Mtfp1^LoxP/LoxP^*
mice were crossed to Alb-Cre recombinase mice to generate LMKO mice. *Mtfp1*
exons 2 and 3 are flanked by LoxP sites (blue arrowheads) and a single FRT site
(green arrowhead), which are excised by Cre-recombinase to yield the deleted (Δ)
*Mtfp1* allele. Created with Biorender.com. **B** PCR genotyping of Mtfp1 alleles from
DNA isolated from indicated different organs of a control and LMKO mice using
allele-specific primers performed once (Supplementary Dataset 5). White adipose
tissue (WAT). **C** Normalized *Mtfp1* expression in liver in control and LMKO male
mice at 24-weeks measured by RNAseq (Supplementary Dataset 1) *n* = 4. Data are
means ± SD, 2-tailed unpaired Student's *t* test. ****p < 0.001. **D** MTFP1, VDAC1 and
TOMM40 protein levels. Equal amounts of protein extracted from livers of male
control (*n* = 4) and LMKO (*n* = 4) mice at 8 weeks of age separated by SDS-PAGE and
immunoblotted with indicated antibodies. **E** Genotype distribution from *Mtfp1^LoxP/LoxP^*
*Alb-Cre^tg/+^* x *Mtfp1^LoxP/LoxP^* intercrosses were not different from expected Mendelian

distributions using two-tailed binomial tests of live female (*n* = 202; *p* > 0.9999) and
male (*n* = 155; *p* = 0.7481) mice. **F** (Top) Representative images of livers from control
and LMKO male mice fed at 24 weeks of age. Scale bar = 1 cm. (Bottom) Repre-
sentative H&E staining *n* = 4–5. Scale bar = 100 μm. **G** Liver mass of 24-week old
control and LMKO male mice fed a normal chow (NCD) *n* = 9. Data are means ± SD.
2-tailed unpaired Student's *t* test, ns = not significant **H** Alanine (ALAT; left) and
Aspartate transaminase (ASAT; right) levels in plasma of control (*n* = 3) and LMKO
(*n* = 3) male mice. Data are means ± SD. 2-tailed unpaired Student's *t* test, ns not
significant. *p < 0.05. **I** Cholesterol (Chol, left) and triglycerides (TG, right) levels in
plasma from **G** ) *n* = 3. Data are means ± SD. 2-tailed unpaired Student's *t* test, ns not
significant. **J** Metabolic cage analyses of NCD-fed control and LMKO male mice at
24 weeks of age. Mean distance covered (interval 3 min), mean food consumption,
and mean water consumption per day during dark and light phases *n* = 3. Data are
means ± SD. 2-tailed unpaired Student's *t* test, ns not significant.

(mitoYFP) showed no differences in fluorescent signal intensity or
surface area (Fig. S2A–C), indicating that mitochondrial mass is unaf-
fected by MTFP1 ablation in primary hepatocytes. Quantification of
mtDNA content in liver tissue revealed no genotype-specific differ-
ences in mice fed a NCD (Fig. S2D) and transmission electron micro-
scopy (TEM) analyses of liver sections showed no differences in
mitochondrial area nor mitochondrial length between control and
LMKO mice (Fig. S2E), further excluding impacts on mitochondrial
mass and morphology. The levels of key proteins regulating mito-
chondrial dynamics remained unchanged in LMKO mice (Fig. S2F) as
did the mitochondrial localization of DRP1, further excluding impacts
on mitochondrial dynamics (Fig. S2G–I). While analysis of primary
hepatocytes revealed a modest increase in mitochondrial elongation in
LMKO mitochondria (Fig. S2C) quantified by supervised machine
learning[24], MTFP1 ablation performed by siRNA-mediated knockdown
in four independent hepatocyte cell lines Huh7.5, HepG2, HC-04 and
Hepa1.6 did not elicit an elongation of the mitochondrial network (Fig.
S3A–D). Finally, RNAseq analyses showed no evidence of gene
expression signatures typically associated with increased mitochon-
drial biogenesis or integrated stress responses (Fig. S1D, Supplemen-
tary Dataset 1) and shotgun liver proteomics performed on control and
LMKO liver extracts revealed no global upregulation of mitochondrial
proteins (Fig. 1D, Supplementary Dataset 2). Taken together, our data
strongly argue that a specific and coordinated increase in both com-
plex IV and V activity in hepatocytes enhances respiration in liver
mitochondria deleted of *Mtfp1*, enabling them to consume nutrient-
derived respiratory substrates at an elevated rate.

## MTFP1 interacts with OXPHOS-related proteins in the liver
To gain insights into the mechanisms responsible for the increased
specific activities, we assessed the relative complex abundance (RCA)[25]
by grouping mitochondrial proteins quantified by proteomics in the
liver of NCD-fed LMKO mice according to the macromolecular com-
plexes to which they belong (Fig. 2H, Supplementary Dataset 2). These
data revealed a significant increase in Complex V subunits, which could
be confirmed by quantitative SDS-PAGE (Fig. 2I) and Blue-native
polyacrylamide gel electrophoresis (BN-PAGE) analyses (Fig. 2J). BN-
PAGE analysis of the steady-state levels of OXPHOS complexes in
LMKO liver mitochondria revealed an increase in Complex V dimers
(Fig. 2J) commensurate with the reported increase in ATPase activity
(Fig. 2F). Taken together, our data suggest that improved assembly
and/or maintenance of Complex V along with an increased activity of
Complex IV is responsible for the enhanced respiration observed in
LMKO liver mitochondria.

To gain insights into the relationship between MTFP1 and macro-
molecular complex assembly in the inner mitochondrial membrane
(IMM), we sought to assess the interactome of MTFP1 in the liver. We
generated a liver-specific transgenic mouse model enabling the expres-
sion of FLAG-MTFP1 from the *Rosa26* locus (henceforth termed

Hepatocyte^FLAG-MTFP1^ mice; Fig. 3A). We verified that the Hepatocyte^FLAG-
MTFP1^ mice, expressed FLAG-MTFP1 correctly in the IMM by protease
protection assay (Fig. 3B). We then subjected liver mitochondria to co-
immunoprecipitation (co-IP) and mass spectrometry (MS/MS) to define
interacting protein partners, which enabled the identification of 112 spe-
cific interactors of MTFP1 with fold change (FC) >2 by MS/MS analysis
(Fig. 3C, D, Supplementary Dataset 3) that could be ascribed to the
mitochondrial proteome according to the Mitocarta 3.0 compendium[26].
We then decided to subdivide this list into binary and enriched inter-
actors: binary interactors were those proteins we could only identify in
Hepatocyte^FLAG-MTFP1^ but not control liver mitochondria CoIP eluates while
enriched interactors were those for which peptide abundance was
greater in CoIP eluates from Hepatocyte^FLAG-MTFP1^ mice (Fig. 3D).

Classification of binary interacting proteins revealed a wide range
of mitochondrial functions, with a conspicuous abundance of factors
involved in mitochondrial translation and OXPHOS (Fig. 3D). On the
other hand, enriched interactors did not reveal an enrichment of
proteins required for mitochondrial translation, but rather those
involved in various metabolic processes within mitochondria, includ-
ing carrier proteins of the SLC25A family. SLC25A22, SLC25A11,
SLC25A20 and SLC25A15 were found to interact with MTFP1 in the liver
(Fig. 3D, Supplementary Dataset 3) yet non-targeted metabolomic
analyses revealed no alterations in the metabolites specifically regu-
lated by these carriers (Fig. S5C-F, Supplementary Dataset 4) and
steady-state levels of SLC25A22, 11, and 20 proteins were not altered in
LMKO mice according to proteomic analyses (Supplementary Data-
set 2). While compelling evidence implicates other carriers such as
SLC25A29, SLC25A47, and SC25A39 in the regulation of OXPHOS
activity[27–29], we found no evidence of their alteration in our OMICs
analyses (Supplementary Dataset 2 and 3). 2D-BN-PAGE analyses
revealed the formation of an MTFP1 macromolecular complex in liver
mitochondria, which was absent in LMKO mice, and that co-migrated
with SLC25A4 (ANT1) (Fig. 3E), as we recently described in the heart[16].
However, we found little overlap with the cardiac interactome of
MTFP1[16] (7 out of 113 proteins), implying that there may be tissue-
specific physical interactions and complex assembly associated with
MTFP1 beyond ANT1 that regulate respiration. Altogether, these data
indicate that post-transcriptional modulation of mitochondrial gene
expression and IMM complex abundance and activities are induced in
a specific manner by the deletion of MTFP1 in hepatocytes, leading to
enhanced mitochondrial respiration.

## *Mtfp1* deletion protects against hepatic steatosis induced by high fat diet
Given enhanced stimulated respiratory activity of LMKO liver mito-
chondria, we next decided to assess the response of LMKO mice to
high fat diet (HFD) feeding, a classical metabolic burden for the liver
that causes nutrient overload, hepatic steatosis, and systemic meta-
bolic dysregulation that can be counteracted by accelerating oxygen

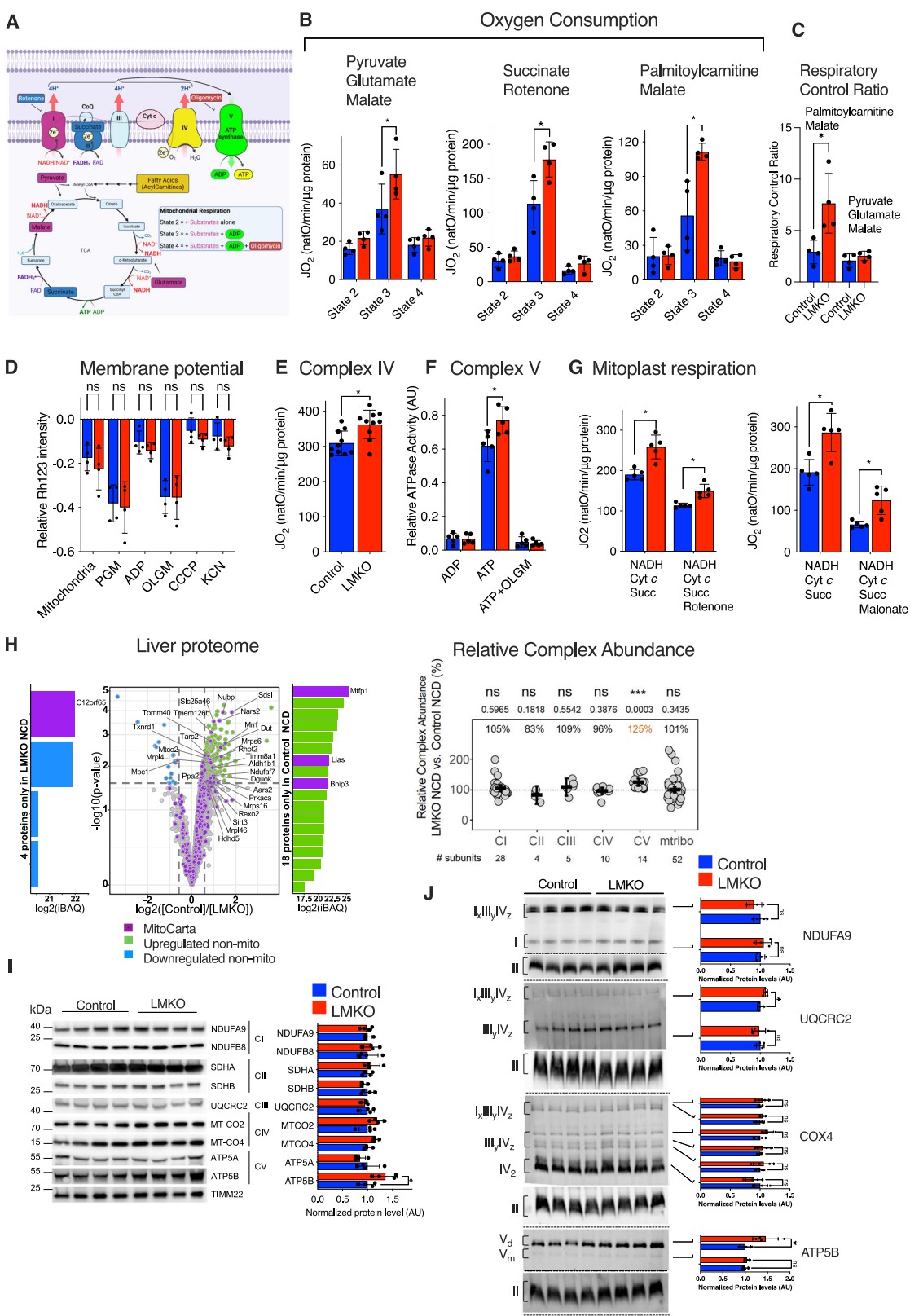

consumption via mitochondrial uncoupling[13,14]. We fed control and LMKO mice a HFD for a period of 16 weeks beginning at 8 weeks of age, which was previously shown to drive diet-induced obesity and MASLD[30]. In control mice, we observed that chronic HFD feeding triggered the hypertrophy of livers (Fig. 4A), which were appreciably paler and heavier (Fig. 4B) than those of control mice on NCD. HFD feeding also induced weight gain (Fig. 4C) and accumulation of hepatic

triglycerides (TG) (Fig. 4D). Histological analyses showed a massive accumulation of lipid droplets, consistent with both micro- and macro-steatosis (Fig. 4E). In contrast, HFD-fed LMKO mice were protected from weight gain (Fig. 4C) and liver dysfunction, exhibiting a 26% of reduction in liver mass compared to diet-matched controls (Fig. 4A, B). Concordantly, histological analyses of livers from HFD-fed LMKO mice revealed a reduction in steatosis (Fig. 4E), a complete rescue of hepatic

**Fig. 2 | Mtfp1 deletion promotes mitochondrial respiration via OXPHOS upregulation. A** Substrates from fatty acid oxidation (mustard) and glycolysis (purple, blue) are metabolized in the TCA cycle which delivers fuels the electron transport chain (ETC) complexes by providing NADH and FADH to complexes I (purple) and II (blue), respectively. ETC creates an electrochemical gradient driving the phosphorylation of ADP by Complex V. Specific inhibitors of complex I (rotenone), complex V (oligomycin). Created with Biorender.com. **B** Oxygen flux (JO$_2$) of isolated liver mitochondria from 16-week old control and LMKO female mice on NCD measured in the presence of pyruvate, glutamate, and malate (state 2) or succinate and rotenone (state 2) or palmitoyl carnitine and malate (state 2) followed by ADP (state 3). Non-phosphorylating respiration was measured in the presence of oligomycin (state 4). Results are represented as the means ± SD of 4 independent experiments. 2-tailed, unpaired Student's $t$ test. *$p < 0.05$. **C** Respiratory Control Ratio (RCR) from 16 week old control and LMKO female mice on NCD measured in the presence of pyruvate, glutamate and malate plus ADP (state 3) or palmitoyl carnitine and malate plus ADP(state 3) divided Non-phosphorylating respiration in the presence of oligomycin (state 4). Respiratory Control Ratio (State3:4) data are represented as the means ± SD of 4 independent experiments. 2-tailed, unpaired Student's $t$ test. *$p < 0.05$. **D** Mitochondrial membrane potential (ΔΨ) measured by quenching of Rhodamine 123 (RH123) fluorescence in liver mitochondria from **B**). ΔΨ was measured in the presence of pyruvate, malate, and glutamate (PGM) (state 2) followed by the addition of ADP (state 3) and Oligomycin (OLGM, state 4), CCCP and potassium cyanide (KCN) $n = 4$. Data represent mean ± SD; 2-tailed Multiple $t$-test. ns not significant. **E** Complex IV activity assessed in isolated liver mitochondria from NCD-fed 16-week old control and LMKO female mice by recording oxygen flux (JO$_2$) in the presence of Antimycin A (AMA), Ascorbate, carbonyl cyanide m-chlorophenylhydrazone (CCCP) and N,N,N',N'-Tetramethyl-p-phenylenediamine dihydrochloride (TMPD), previously incubated with of pyruvate, glutamate, and

malate (PGM) and ADP $n = 10$. Data are means ± SD. 2-tailed, unpaired Student's $t$ test. *$p < 0.05$. **F** Complex V (ATP synthase) activity assessed in isolated liver mitochondria from NCD-fed 16-week old control and LMKO female mice in the reverse mode in the presence of ATP. Oligomycin (OLGM) or ADP were used as negative controls $n = 5$. Data are means ± SD. 2-tailed, unpaired Student's $t$ test. *$p < 0.05$. **G** Mitoplast respiration assessment. Mitoplasts were supplied with either NADH, cytochrome $c$, succinate and malonate, (complex I) or NADH, cytochrome $c$, succinate and rotenone (complex II) $n = 5$. Data are means ± SD. 2-tailed, unpaired Student's $t$ test. *$p < 0.05$. **H** (Top) Volcano plot of liver proteome of LMKO male mice analyzed by mass spectrometry.(Purple) Mitochondrial proteins (MitoCarta 3.0), (Green) Non-mitochondrial proteins more abundant in control liver. (Blue) Non-mitochondrial proteins significantly more abundant in LMKO liver. (Bottom) Relative complex abundance (RCA) plot comparing the levels of the OXPHOS complexes (CI-CV) and the mitoribosome (mtribo) between LMKO and control mice fed. NCD. The graph represents the relative values of each complex ratio between LMKO and controls. The dotted line represents the control mean value of each complex and error bars represent 95% confidence interval of the mean. $n = 4$. 2-tailed Paired $t$-test. ***=$p < 0.001$, ns=non-significant. **I** Proteins levels of different OXPHOS complexes subunits. Equal amounts of protein extracted from hepatic mitochondria isolated from NCD-fed 16-week old control ($n = 4$) and LMKO ($n = 4$) male mice were separated by SDS-PAGE, immunoblotted with indicated antibodies (left) and quantified by densitometry (right) relative to TIMM22 loading control. Data are means ± SD. 2-tailed, unpaired Student's $t$ test. *$p < 0.05$. ns not significant. **J** 1D BN-PAGE (left) and quantified by densitometry (right) relative to SDHA loading control of OXPHOS complexes in hepatic mitochondria isolated from NCD-fed 16-weeks old female control and LMKO. $n = 4$. Data are means ± SD. 2-tailed, unpaired Student's $t$ test. *$p < 0.05$. ns not significant.

TG accumulation back to levels measured in NCD-treated mice (Fig. 4D), and a reduction in circulating levels of ALAT by 64% and ASAT by 32% (Fig. 4F). Metabolic protection against HFD-feeding in LMKO mice was reflected at the molecular level: liver RNAseq analyses showed that HFD feeding induced significant transcriptional dysregulation in control livers with 854 upregulated genes and 758 downregulated genes (Figs. 4G, S5). Gene ontology analyses revealed an enrichment of various metabolic pathways dysregulated by HFD-feeding in control mice including, lipid and branched chain amino acid metabolism, PPAR signaling, and steroid biosynthesis, and fatty acid metabolism (Fig. 4H). These alterations were absent in HFD-fed LMKO mice, in which only 214 upregulated genes and 164 downregulated genes were detected, most of which were involved in pathways of stress response and immune signaling (Fig. 4H). At the protein level, we observed that in HFD feeding in control mice induced global reduction in complex IV by liver proteomic analyses (Fig. S4A). This response was blunted in LMKO mice (Fig. S4B), leading to a relative increase in complex IV proteins in HFD-fed LMKO mice compared with littermate, diet-matched controls (Fig. S4C). Taken together, our OMICs studies revealed a global protection of LMKO mice against diet-induced hepatic remodeling of biosynthetic and signaling pathways that intersect at mitochondria.

Given the protection against hepatocyte and liver damage observed in vivo and in vitro upon the deletion of *Mtfp1*, we asked whether the protection against HFD-induced metabolic dysregulation of the liver could be explained by differential sensitivity to cell death. However, TUNEL assays performed on histological sections from HFD-fed control and LMKO mice revealed an absence hepatocyte apoptosis (Fig. 4I), which is consistent with previous findings that HFD causes limited liver cell death[30]. Altogether, our data demonstrate that *Mtfp1* deletion in hepatocytes confers metabolic resistance to hepatic steatosis in vivo in a manner that is independent of apoptotic resistance.

### *Mtfp1* deletion improves hepatocyte metabolism in a cell-autonomous manner
Having demonstrated that LMKO mice are protected against diet-induced steatosis, we next sought to define the breadth of systemic

protection by performing a battery of metabolic tests on control and LMKO mice after HFD-feeding. Fat accumulation is known to affect systemic glucose homeostasis. Consistent with improved liver metabolism promoted by the ablation of MTFP1 in hepatocytes, metabolic cage analyses revealed improved systemic respiration in LMKO mice fed an HFD. The lower respiratory exchange ratio (RER) observed in LMKO mice (Fig. 5A) suggests that LMKO mice metabolize nutrient-derived lipids more efficiently than control littermates, which can very likely be attributed to intrinsic differences in the livers of these mice. Food and water intake were not altered between control and LMKO mice nor were distance measurements on HFD (Fig. S6A–C), pointing to a specific effect of MTFP1 on energy expenditure by the liver that is revealed under HFD feeding. Metabolic dysregulation caused by HFD feeding is known to increase de novo glucose synthesis by the liver of rodents[31] and so we sought to assess gluconeogenesis by performing intraperitoneal pyruvate tolerance tests (IP-PTT). We observed a 1.5-fold increase blood glucose levels in HFD-fed control mice relative to NCD littermates, which was reduced by 14% in the HFD-fed LMKO group (Fig. 5B), indicating that *Mtfp1* deletion in hepatocytes protects the liver against diet-induced dysregulation of gluconeogenesis. Consistent with these findings, intraperitoneal glucose tolerance tests (IP-GTT) revealed LMKO mice to be modestly protected from HFD-induced glucose intolerance. In control mice, HFD feeding led to a 1.7-fold increase of the area under curve (AUC) of IP-GTT relative to normal chow diet (NCD) controls, which reduced by 17% in HFD-fed LMKO mice relative to diet-matched controls (Fig. 5C). To exclude that improved glucose tolerance in HFD-fed LMKO mice was the consequence of altered insulin resistance, rather than improved gluconeogenesis, we performed intraperitoneal insulin tolerance tests (IP-ITT). We observed no genotype-specific differences in insulin sensitivity (Fig. 5D) on either HFD or NCD, indicating that improved glucose tolerance in LMKO mice is not caused by increased systemic insulin sensitivity. Concordantly, we did not observe differences in basal insulin levels between control and LMKO mice fed a NCD (Fig S7D) and no differences in hepatic insulin signaling of fasted LMKO mice stimulated with insulin (Fig. S7, F). While LMKO mice gained less weight upon HFD feeding (Fig. 4C), we observed no genotype-specific

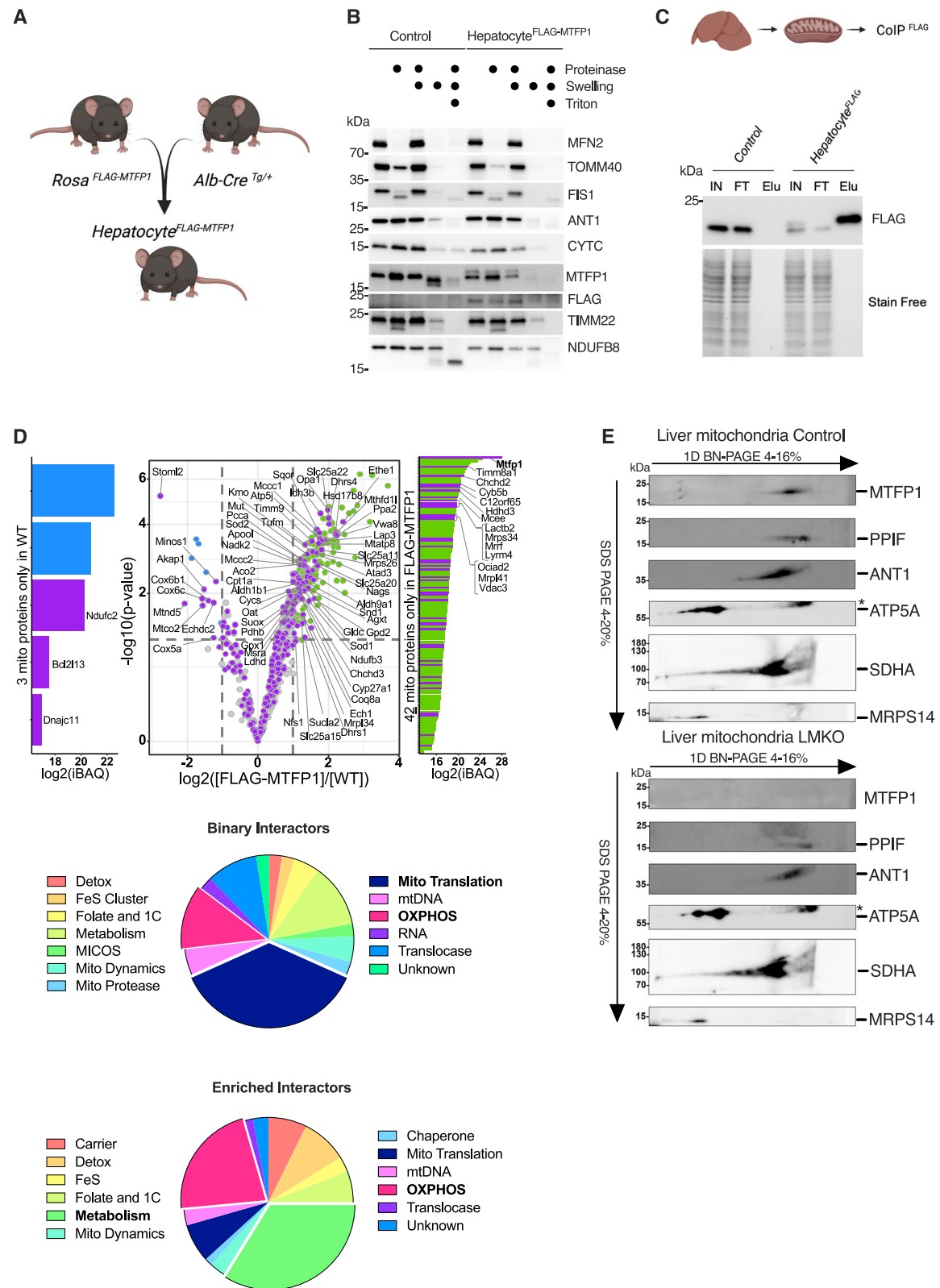

**Binary Interactors**

Detox, FeS Cluster, Folate and 1C, Metabolism, MICOS, Mito Dynamics, Mito Protease, **Mito Translation**, mtDNA, **OXPHOS**, RNA, Translocase, Unknown

**Enriched Interactors**

Carrier, Detox, FeS, Folate and 1C, **Metabolism**, Mito Dynamics, Chaperone, Mito Translation, mtDNA, **OXPHOS**, Translocase, Unknown

differences in body mass composition by NMR minispec analysis (Fig. 5E) nor in the circulating levels of cholesterol and triglycerides (Fig. 5F). In line with these findings, LMKO mice were not protected from extra-hepatic fat mass accumulation (Fig. 5G), white adipose tissue accumulation nor adipocyte hypertrophy (Fig. S7G, H), all of which are common metabolic consequences of diet-induced obesity[32]. We also did not observe differences *Ucp1* nor *Prdm16* expression in

brown adipose tissue from LMKO mice (Fig S7I, J), indicating that BAT function is unlikely to be indirectly affected by liver-specific deletion of *Mtfp1*. Altogether, our data demonstrate that the protection against diet-induced metabolic dysregulation conferred to mice by liver-specific deletion of *Mtfp1* is primarily restricted to the liver.

To determine whether hepatocyte deletion of *Mtfp1* prevents steatosis in a cell-autonomous manner, we established an assay in

**Fig. 3 | MTFP1 interacts with various IMM components in the liver. A** Generation of transgenic (Tg) Hepatocyte[FLAG-MTFP1] male mice constitutively expressing FLAG-MTFP1 from the *Ros26* locus. Created with Biorender.com. **B** Protease protection assay of liver mitochondria isolated from control and transgenic (Tg) Hepatocyte[FLAG-MTFP1] male mice analyzed by immunoblot with indicated antibodies performed once. **C** Liver mitochondria co-immunoprecipitation (co-IP) in WT and FLAG_MTFP1 mitochondria. Elution fraction (Elu) performed thrice with similar results. Stain-free used as a loading control. Schematic created with Biorender.com **D** Volcano plot of the FLAG-MTFP1 interactome analyzed by mass spectrometry (left). (Purple) Mitochondrial proteins exclusively present in FLAG-MTFP1 eluates (binary interactors) or enriched greater than two-fold listed in Supplementary Data 5. (Green) Non-mitochondrial proteins enriched in Hepatocyte[FLAG-MTFP1] livers.

On the y-axis, the $-\log10$(p-value) are displayed for all proteins with p-values resulting from LIMMA moderated t-tests. Dashed lines are the thresholds used to claim that a protein is significantly differentially abundant between the biological conditions (absolute log2 fold-change superior to one and adjusted p-value below 1% FDR after applying an adaptive Benjaminin-Hochberg procedure). (Blue) Non-mitochondrial proteins enriched in control liver. (Right) Functional classification of 113 mitochondrial proteins identified in Co-IP eluates (Supplementary Dataset 3). **E** Second dimension electrophoresis (2D BN-PAGE) analysis of hepatic mitochondria isolated from control (top) and LMKO (bottom) female mice fed a NCD immunoblotted using the indicated antibodies performed twice with similar results. *=nonspecificity.

which we could mimic HFD-induced steatosis in primary hepatocytes isolated from NCD-fed mice. Primary hepatocytes were plated and cultured in the presence of Intralipid (IntLip): a complex lipid emulsion composed of linoleic, oleic, palmitic, and stearic acids, which are the most abundant fatty acids found in HFD. We optimized our assay conditions to ensure that limited damage, death, or differentiation under both treated and non-treated (NT) conditions was occurring. Indeed, IntLip feeding of primary hepatocytes isolated from NCD-fed control mice led to a 1.6-fold increase in intracellular lipid accumulation after 24 h, which could be visualized by live-cell imaging with BODIPY fluorescence (Fig. 5H) and quantified by high-throughput confocal imaging (Fig. 5I). Notably, deletion of *Mtfp1* in primary hepatocytes rescued this phenotype: BODIPY staining intensities in NT and IntLip-treated *Mtfp1*[−/−] cells were indistinguishable to those of NT primary hepatocytes isolated from control littermates. Thus, we conclude that the deletion of *Mtfp1* in hepatocytes confers direct protection to the liver against HFD-induced metabolic dysregulation by improving hepatocyte metabolism in a cell-autonomous manner.

### *Mtfp1* deletion protects against hepatic cell death

Given the previous implications of MTFP1 in cell death sensitivity reported in various cultured cell lines[18,20,33] and the association of MTFP1 and liver tumor progression in humans[19], we sought to directly test whether MTFP1 was involved in the regulation of cell survival in vivo. We measured the response to liver apoptosis by treating 3-month old mice for 24 h with FAS antibodies, which specifically induces apoptotic liver damage via the activation of the FADD signaling pathway in a manner than is regulated by IMM integrity[34]. Macroscopically, the livers of WT mice treated with FAS showed hepatic damage characterized by the appearance of fibrotic and necrotic foci (Fig. 6A). FAS-induced liver damage resulted in a loss of liver mass (Fig. S8A), a dramatic increase of ALAT, ASAT and lactate dehydrogenase (LDH), a more general marker of tissue damage, by 35, 85 and 45-fold respectively in control livers (Fig. 6B, S8B). Hematoxylin and eosin (H&E) staining of control livers revealed tissue disruption, characterized by the presence of apoptotic bodies and hyper-eosinophilic cytoplasm, conferring large areas of architecture dysregulation (Fig. 6C), all of which features are consistent with apoptotic liver damage[35]. In contrast, H&E analyses of LMKO livers revealed far less FAS-induced damage (Fig. 6C) and plasma analyses revealed an 84% reduction of ALAT, an 86% reduction of ASAT, and a 97% reduction of LDH compared to FAS-injected control mice (Fig. 6B, Fig. S8B). Finally, TUNEL assays performed on histological liver sections demonstrated a profound reduction of cell death in FAS-treated LMKO mice compared to control mice (Fig. 6D). Taken together, these studies demonstrate that liver-specific *Mtfp1* depletion protects against apoptosis-induced liver damage in vivo.

To understand whether MTFP1 ablation protects hepatocytes from cell death in a cell-autonomous manner, we isolated primary hepatocytes from control and LMKO mice at 8–10 weeks of age and performed quantitative cell death assays ex vivo. To induce cell death, primary hepatocytes were treated with hydrogen peroxide ($H_2O_2$)

[1mM] at 16 h post-plating and cell death was by kinetically monitored by propidium iodide (PI) uptake using an automated live-cell imaging approach powered by supervised machine learning previously developed for cultured cell lines[16,24]. Consistent with our observations in vivo, *Mtfp1*[−/−] primary hepatocytes treated with $H_2O_2$ ex vivo showed a significant resistance against cell death (52% reduction at 4 h) compared to control primary hepatocytes (Fig. 6E, F), arguing that MTFP1 ablation confers protection to autonomous hepatocyte death ex vivo. Cell death in the liver, and more specifically in hepatocytes, can be tightly regulated through the opening of the mitochondrial permeability transition pore (mPTP)[36,37]. To test whether *Mtfp1* deletion impacted mPTP opening, we isolated hepatic mitochondria and simulated mPTP opening with calcium chloride overload (Fig. 6G). We observed a reduced sensitivity of mPTP opening in hepatic mitochondria isolated from LMKO mice compared to control littermates (Fig. 6G), which could be rescued by the pre-treatment with the mPTP inhibitor Cyclosporin A (CsA)[37]. Taken together, we conclude that *Mtfp1* deletion in hepatocytes confers protection against mPTP opening and cell death induction.

### Discussion

Pre-clinical transgenic animal models have been instrumental in dissecting the underlying molecular remodeling of hepatocytes during the development of fatty liver disease and its sequelae. These studies have demonstrated the unassailable pathophysiological relevance of mitochondria, yet data from a large array knockout mice of mitochondrial genes have yielded a more complex picture: some liver-specific knockouts confer resistance to MASLD and MASH while others inhibit normal liver function[38–42]. Pleotropic effects of the individual mitochondrial genes that were targeted in previous studies combined with the potential for metabolic crosstalk between tissues in vivo has made it challenging to pinpoint how modulating mitochondrial activity may be used to combat liver disease. Here, we discovered that Mitochondrial Fission Process 1 (MTFP1) plays an important role metabolic role in the liver of mice that is critical for MASLD but not under basal conditions. MTFP1 was first identified as a metabolically-regulated inner mitochondrial membrane (IMM) protein[20] initially implicated in mitochondrial fission and cell death resilience in a variety of cell lines[17–19,43]. In the liver, the discovery that MTFP1 protein expression is predictive of hepatocellular carcinoma survival and recurrence risk in humans[19], prompted us to directly investigate its role in the liver.

Here, we created Liver-specific *Mtfp1* knockout (LMKO) mice (Fig. 1A–E, S1A, B) by expressing a liver-specific Cre-recombinase[44] in a conditional mouse model of *Mtfp1*[16]. We discovered that LMKO mice are healthy and virtually indistinguishable from littermate control mice, showing no defects in liver function and no indication of liver damage nor inflammatory responses and liver RNAseq analyses only showed 0.1% of genes were dysregulated, none of which were involved in pathogenic hepatic remodeling (Supplementary Dataset 1). While the livers of LMKO mice appeared unchanged, characterization of hepatic mitochondria revealed two major functional improvements: increased

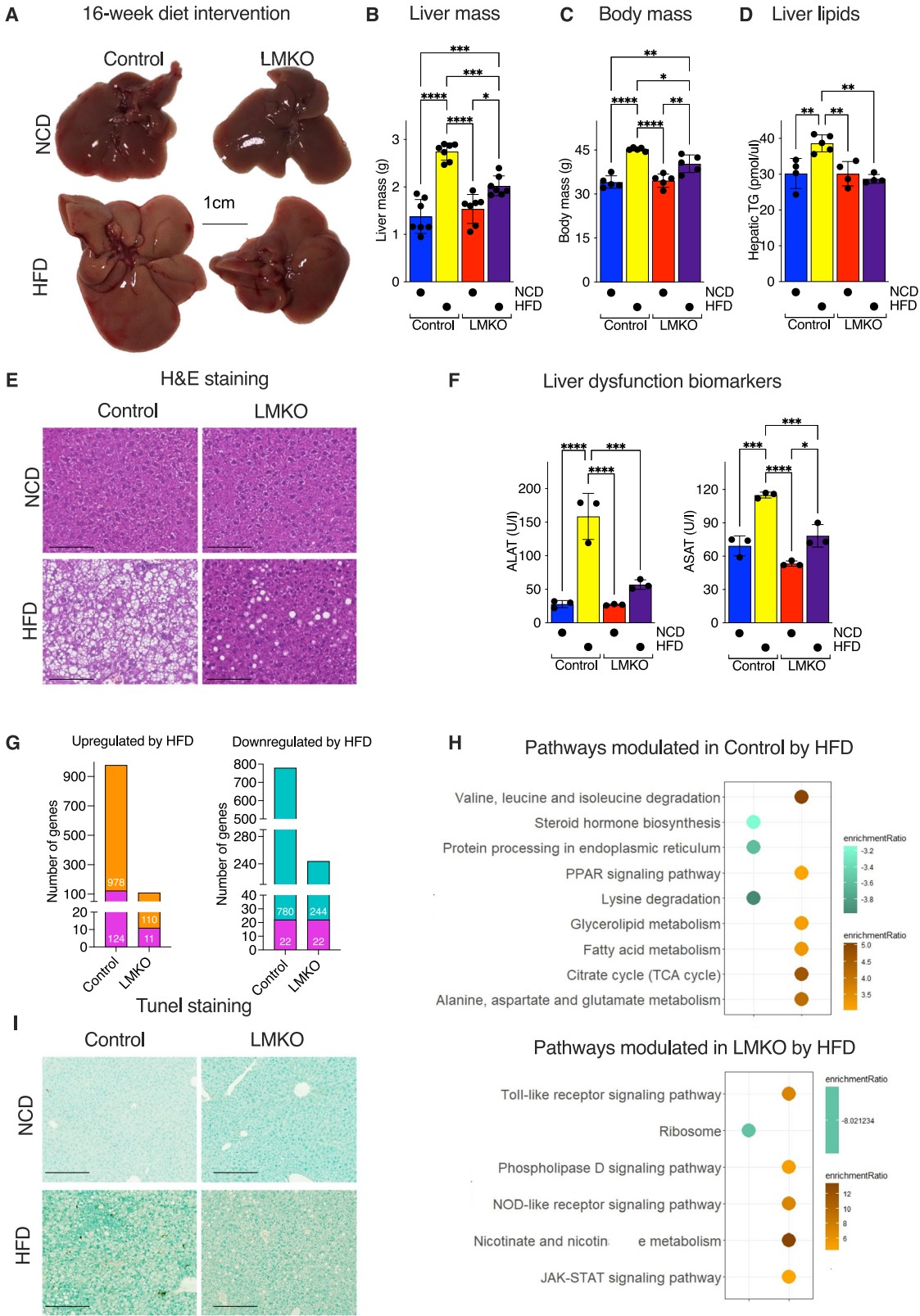

respiratory capacity and resistance to permeability transition pore opening. High resolution fluor-respirometry measurements of MTFP1-deficient liver mitochondria showed increased ADP-stimulated (state 3) oxygen consumption rates under all substrates we tested, which was associated a commensurate increase in the specific activities of complex IV and complex V (Fig. 2B–F). Quantitative label-free proteomic studies revealed a ratio-metric increase in the levels of Complex V

subunits, which paralleled the increased levels of assembled Complex V detected by BN-PAGE, pointing to an increased assembly and/or maintenance of the ATP synthase complex (Fig. 2H–J). Enhanced respiration was not associated with proton leak (Fig. 2B, D) nor cristae alterations (Fig. S2E) that we reported in MTFP1-deficient hearts[16], leaving us wondering: how does MTFP1 ablation enhance OXPHOS activity and mitochondrial coupling and do so specifically in the liver?

**Fig. 4 | Hepatic deletion of *Mtfp1* protects mice from high fat diet-induced steatosis. A** Representative images of livers from control and LMKO male mice fed a normal chow diet (NCD) or a high fat diet (HFD) for 16 weeks. Scale bar = 1 cm. **B** Liver mass of 24-week old control and LMKO male mice fed a NCD or HFD for 16 weeks. *n* = 7. Data are means ± SD. 2-way ANOVA, *$*p < 0.05$, ***$p < 0.001$, ****$p < 0.0001$. **C** Body mass of control and LMKO male mice fed a NCD or HFD for 16 weeks. *n* = 7. Data are means ± SD. 2-way ANOVA, *$*p < 0.05$, **$p < 0.01$, ****$p < 0.0001$. **D** Triglycerides (TG) measurement in liver tissue from 24-week old control and LMKO male mice fed a NCD or HFD for 16 weeks. Control NCD *n* = 4, Control HFD *n* = 5, LMKO NCD *n* = 4, LMKO HFD *n* = 4. Data are means ± SD. 2-way ANOVA, **$p < 0.01$, **E** H&E staining of control and LMKO livers from **A**) performed twice in NCD and thrice in HFD with similar results. Scale bar = 100 μm. **F** Alanine (ALAT; left) and Aspartate transaminase (ASAT; right) levels in plasma of 24-week old control and LMKO male mice fed a NCD or HFD for 16 weeks. *n* = 4. Data are means ± SD. 2-way ANOVA, *$*p < 0.05$, ***$p < 0.001$, ****$p < 0.0001$. **G** Differential modulation of liver gene expression by HFD. Upregulation (left) and down-regulation (right) of mitochondrial (pink) and non-mitochondrial genes (orange or teal) in LMKO and control livers (Supplementary Dataset 1). **H** Pathway enrichment analysis of differentially expressed genes depicted in **G**). **I** TUNEL staining of control and LMKO livers from **A**) performed once in NCD and thrice in HFD with similar results. Scale bar=100μm.

The spare respiratory capacity of hepatocytes has been reported to supersede that of cardiomyocytes, indicating that under basal conditions cardiomyocyte mitochondria are working far closer to their maximal capacity[45]. In contrast, morphometric studies of cardiac mitochondria from rodents reveal cristae densities that approach the physical and functional limits exemplified by tuna[46] and hummingbird hearts[47], consistent with the notion that there may be little additional IMM real estate upon which to build additional OXPHOS complexes in the heart[48]. In the liver, mitochondria have a comparatively reduced IMM density, OXPHOS protein content[49], and IMM protein half-lives[50] and would, therefore, be capable of increasing OXPHOS complex assembly to enhance mitochondrial coupling and respiration. In rodents, systemic thyroid hormone injections can increase hepatocyte cristae density to increase the content and activity of the respiratory chain[51,52]. On the other hand, genetic models of obesity manifest the oppositive effect, with altered cristae organization in swollen mitochondria and reduced levels of cytochrome *c* oxidase[53], which we also confirmed by proteomic studies of HFD-fed control mice (Fig. S4A, B). Yet to the best of our knowledge, LMKO mice represent the first genetic mouse model that elevates respiration by increasing the activity of Complex IV and V rather than by boosting total mitochondrial mass or by promoting uncoupling (Fig. 7).

MTFP1 ablation in the liver enhances state 3 respiration regardless of the substrate source (Fig. 2B) yet this increase is most profound when the palmitoyl-carnitine is supplied in excess, as evidenced by the increased respiratory control ration (RCR) in the presence of fatty acids but not carbohydrate-derived substrates (Fig. 2C), revealing that MTFP1-deficient liver mitochondria can couple voracious beta oxidation to enhanced OXPHOS activity. A clear prediction of increased beta oxidation capacity of hepatocyte mitochondria is an enhanced propensity to metabolize excess lipids and therefore a resistance against diet-induced hepatic steatosis in vivo[2]. Indeed, when we challenged LMKO mice with a long-term a high fat diet (HFD), they showed remarkable resistance to fatty liver disease and metabolic dysregulation despite equal levels of food consumption and in-cage activity. Metabolic protection appears to be restricted to the liver, as LMKO mice were not protected against the increased extra-hepatic adiposity, dysregulated systemic glucose metabolism, altered white and brown adipose tissue status, nor aberrant glucose-stimulated insulin secretion by the pancreas that caused by HFD feeding. Mimicking diet-induced steatosis in the tissue culture dish with primary hepatocytes isolated from NCD-fed mice, we could show that excessive lipid droplet accumulation was blunted in *Mtfp1*$^{-/-}$ primary hepatocytes ex vivo that is consistent with enhanced mitochondrial coupling in these cells (Fig. 5H).

The discovery that MTFP1 ablation in the liver is benign and even beneficial under stress conditions is in stark contrast to our recent report that *Mtfp1* deletion in cardiomyocytes causes heart failure[16]. However, defects in inner membrane integrity observed in *Mtfp1*$^{-/-}$ cardiac mitochondria were not observed in *Mtfp1*$^{-/-}$ liver mitochondria: membrane potential was maintained (Fig. 2D) and mPTP was increased rather than decreased (Fig. 6G). Moreover, the livers of LMKO mice show increased mitochondrial respiratory capacity (Fig. 2B) and no

signs of cell death or inflammation (Fig. 4I.), while MTFP1 deficiency in the heart triggered cardiomyocyte death, fibrotic remodeling, and heart failure[16]. At first, we found these physiological discrepancies perplexing, since the heart and liver are both rely heavily on mitochondria yet further exploration of the literature provided conspicuous examples demonstrating that these two organs can in fact respond differently to mitochondrial dysfunction. For example, ablation of Complex I activity in the cardiomyocytes can cause a fatal hypertrophic cardiomyopathy in mice[54] while its ablation in hepatocytes does not impact liver structure and function[42]. Similarly, interfering with the activity of the mitochondrial fusion proteins OPA1[55,56] or Mitofusins[57] specifically in cardiomyocytes yields disastrous outcomes for cardiac function, yet in doing so in the liver is either benign[38,39] or even advantageous against MASH[40]. Similarly, cardiac deletion of the adenine nucleotide transporter elicits cardiac hypertrophy but in the liver, its genetic or biochemical inhibition in the liver can be beneficial, protecting against hepatic steatosis and metabolic dysregulation[41,58]. In mice, system-wide disruption of mitochondrial genes such as *Chchd10*[59,60], *Dnm1l*[61], *Wars2*[62], and *Cox1*[63] can trigger multi-organ dysfunction in which the liver, but not the heart, is spared, further illustrating that the liver may respond differently (or not at all) to genetic insults otherwise catastrophic to other tissues.

But why does the ablation of MTFP1 impact liver mitochondria so differently? It is likely that divergent cell type-specific effects reflect the different metabolic, structural, and functional characteristics of mitochondria they harbor. Indeed, compelling differences between the heart and liver have been documented at the level of mitochondrial network morphology, IMM and cristae organization[64], mitochondrial proteome composition and protein half-lives[50], OXPHOS complex organization and assembly[65], substrate utilization and enzymatic activities of the respiratory chain[65,66], and even the molecular regulation of the mPTP[36,37]. Consistent with these observations, we found little overlap between the differentially expressed mitochondrial genes and proteins of knockout hearts and livers: no genes besides *Mtfp1* were commonly dysregulated between both knockout tissues while only 13% (35 out of 264 proteins) shared common differential expression patterns (Fig. S9A, B). MTFP1 interactomes were also remarkably dissimilar, with only 7 out of 113 interacting partners of liver MTFP1 found to overlap with the cardiac MTFP1 interactome. MTFP1 is devoid of conserved motifs or catalytic domains and forms a complex of approximately 150 kDa, which is significantly smaller than the fully assembled mito-ribosome and ATP synthase complexes (Fig. 3E) with whose subunits MTFP1 interacts and thus we posit that the impacts on mitochondrial function that result from the ablation MTFP1 may reflect its role as a protein scaffold in the IMM. To explore the functional relevance of MTFP1 interactions, we intersected the interactomics data from the Hepatocyte$^{FLAG-MTFP1}$ mice with the differentially expressed proteins (DEPs) in LMKO livers and identified 5 overlapping bona fide mitochondrial proteins: MRRF, TIMM8A1, RARS2, MRPL43, and ALDH1B1. MRRF (2.5-fold decrease), RARS2 (1.7-fold increase) and MRPL43 (1.55-fold increase) are involved in mitochondrial translation, however, RCA analyses in LMKO livers showed no effect on mito-ribosome assembly (Fig. 2H) and the steady-state

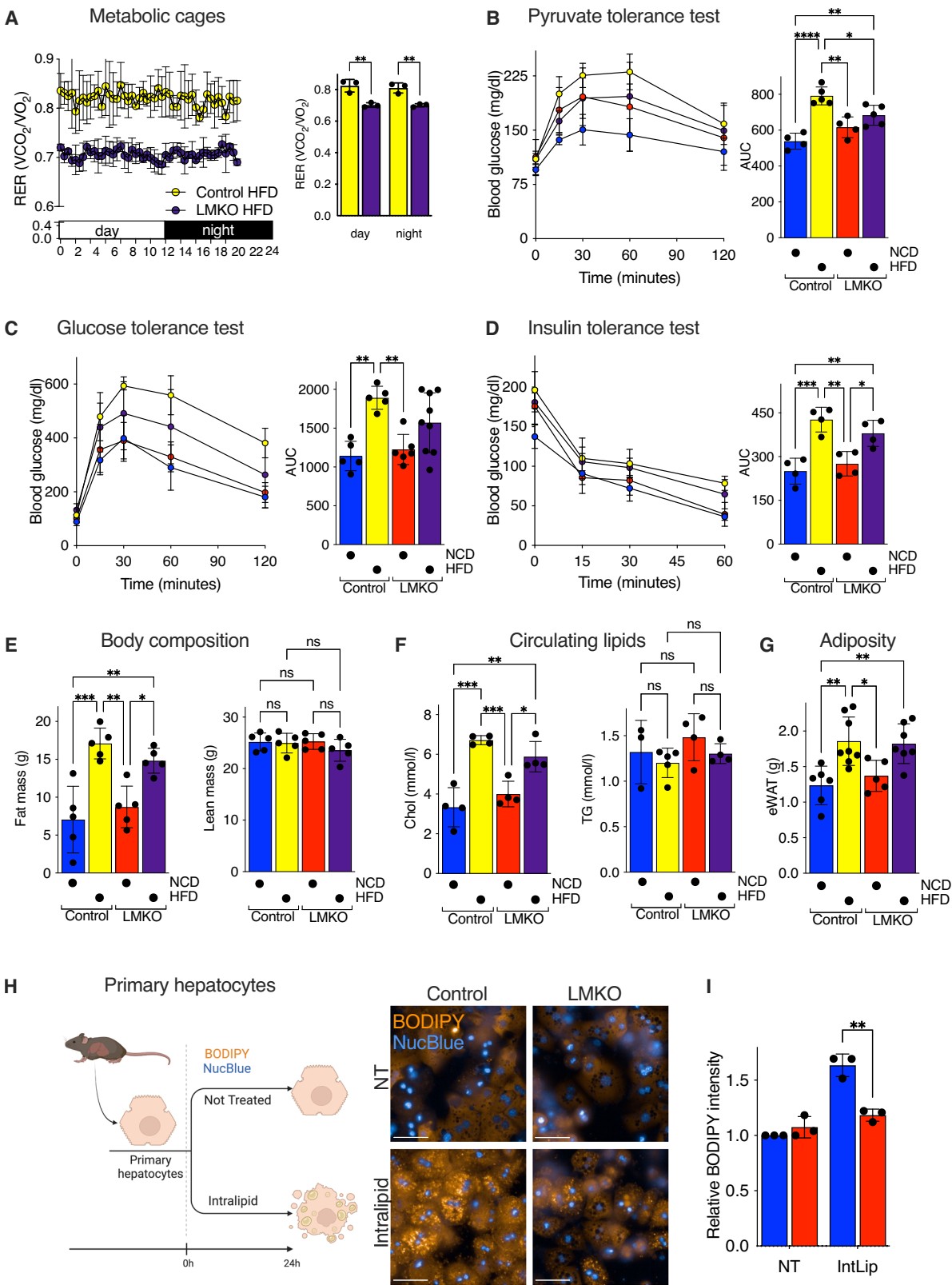

levels of mtDNA-encoded proteins were not globally dysregulated (Supplementary Dataset 2), indicating that these changes have little functional impact. TIMM8A1, which is increased by 1.9-fold in LMKO livers, is involved in the import and insertion of hydrophobic IMM proteins, however, no known associations with Complex V have been reported[67]. ALDH1B1, upregulated by 1.9-fold in LMKO livers, is a mitochondrial enzyme involved in ethanol detoxification with no

established links to OXPHOS activity[68]. Therefore, future studies are required to interrogate the functional relevance of these changes in LMKO mice.

MTFP1 co-migrates with both adenine nucleotide translocase (ANT1) and Cyclophilin D (encoded by *Ppif*), both of which have been previously physically and functionally connected to mitochondrial ATP synthesis[69–73] and the mPTP. Our data also support a role for

**Fig. 5 | LMKO mice are protected against diet-induced metabolic dysregulation. A** Whole-body Respiratory Exchange ratios (RER; $VCO_2/VO_2$) in control and LMKO male mice after 16 weeks of HFD. $n = 3$. Data are means ± SD. 2-way ANOVA, $**p < 0.01$. **B** Pyruvate tolerance tests performed after 18h fasting. Left panel: glycaemia measured at indicated time. Right panel: area under the curve (AUC). $n = 4$ NCD; $n = 5$ HFD groups. Data are means ± SD. 2-way ANOVA, $*p < 0.05$, $**p < 0.01$, $****p < 0.0001$. **C** Glucose tolerance tests performed after 18h fasting. Left panel: glycaemia measured at indicated time. Right panel: area under the curve (AUC). Control NCD $n = 4$, Control HFD $n = 5$, LMKO NCD $n = 6$, LMKO HFD $n = 9$. Data are means ± SD 2-way ANOVA, $**p < 0.01$. **D** Insulin tolerance tests performed after 6 h fasting. Left panel: glycaemia measured at indicated time after insulin injection. Right panel: area under the curve (AUC). $n = 4$. Data are means ± SD. 2-way ANOVA, $*p < 0.05$, $**p < 0.01$, $***p < 0.001$. **E** Body composition analysis of control and LMKO male mice fed NCD or HFD by Nuclear Magnetic Resonance (NMR). $n = 5$. Left panel: fat mass; right panel: lean mass. Data are means ± SD. 2-way ANOVA, $*p < 0.05$, $**p < 0.01$, $***p < 0.001$, ns not significant **F** Plasma cholesterol (Chol, left) and triglycerides (TG, right) in control and LMKO male mice fed NCD or HFD. For cholesterol: Control NCD $n = 4$, Control HFD $n = 4$, LMKO NCD $n = 4$, LMKO HFD $n = 4$. For triglycerides: Control NCD $n = 3$, Control HFD $n = 5$, LMKO NCD $n = 4$, LMKO HFD $n = 4$. Data are means ± SD. 2-way ANOVA, $*p < 0.05$, $**p < 0.01$, $***p < 0.001$, ns not significant **G** Epididymal white adipose tissue (eWAT) mass in control and LMKO male mice fed NCD or HFD $n = 5$ NCD, $n = 7$ HFD. Data are means ± SD. 2-way ANOVA, $*p < 0.05$, $**p < 0.01$. **H** Ex vivo steatosis measurement (Created with Biorender.com). Representative images of primary hepatocytes isolated from control and LMKO male mice treated with IntraLipid [0.65%] for 24h and stained for nuclei (NucBlue, blue) and lipid droplets (BODIPY, orange) performed thrice with similar results. **I** BODIPY intensity quantification of **H**) Data are means of 3 independent experiments ± SD. 2-tailed unpaired Student's $t$ test, $**p < 0.01$. Scale bar = 50 μm.

MTFP1 as a regulator of mPTP activity in the liver, as inactivation of hepatocytes slows mPTP opening response in liver mitochondria and protects against liver cell death in primary hepatocytes and in mice (Fig. 6). This pro-survival role of MTFP1 in the liver appears to be independent from the metabolic protection its deletion confers against HFD-induced hepatic steatosis, since HFD-feeding does not cause cell death (Fig. 4I).

The metabolic protection conferred to MTFP1-deficient livers that are already hyper-proficient in mitochondrial respiration and fatty acid oxidation is best appreciated under chronic HFD feeding (Fig. 7). Intriguingly, recombinant inbred mouse models that are susceptible to MASLD/MASH show a downregulation of *Mtfp1* expression in the liver while those that are resistant do not[74]. While diet-induced hepatic steatosis predisposes humans to MASH and cirrhosis, HFD-feeding alone does trigger MASH in laboratory mice so we have not yet learned whether LMKO mice are protected from the downstream sequelae of hepatic steatosis. Nevertheless, LMKO mice have availed themselves to be a unique tool to understand how enhancing mitochondrial respiration can combat metabolic liver disease characterized by hepatic steatosis. In humans, from 20% to 27% of individuals manifesting steatosis will progress to steatohepatitis and of those, 10% will develop cirrhosis and, eventually hepatocellular carcinoma[75,76]. Defining the contribution of mitochondrial dysfunction(s) in the progression from simple steatosis to hepatocellular carcinoma has been challenging given the interdependence of the various signaling and metabolic functions of mitochondria. Here, our study provides a tractable genetic target able to enhance respiration and limit free fatty acid-induced hepatic steatosis independently of promoting mitochondrial biogenesis and without uncoupling mitochondria. Hence, the mitochondrial mechanisms through which MTFP1 ablation protects against hepatic steatosis appears to be functionally distinct from pathways currently being pursued as therapeutic interventions for MASLD and associated pathologies[5,77,78]. Currently, there is limited information on the beneficial effects in the liver of boosting both Complex IV and V activities concomitantly and further studies are currently underway to determine the breadth of protection that MTFP1 ablation in hepatocytes confers in the context of NASH and may prove useful for the in vivo exploration and development of therapeutic targets for other inherited and acquired liver diseases.

## Methods

### Animals

Animals were handled according to the European legislation for animal welfare (Directive 2010/63/EU). All animal experiments were performed according to French legislation in compliance with the European Communities Council Directives (2010/63/UE, French Law 2013-118, February 6, 2013) and those of Institut Pasteur Animal Care Committees (CETEA is Comité d'Ethique en Expérimentation Animale 89). The specific approved protocol number is #20687-

2019051617105572. Mice were housed within a specific pathogen free facility and maintained under standard housing conditions of a 14–10 h light-dark cycle, 50–70% humidity, 19–21 °C with free access to food and water in cages enriched with bedding material and gnawing sticks. Alb-Cre recombinase[44] and mitoYFP[79] mouse lines are commercially available (MGI: 2176946 and 5292496). The *Mtfp1* conditional mouse model[16] was generated by PolyGene on a C57Bl/6N genetic background using two LoxP sites and an FRT-flanked neomycin cassette. The first LoxP site and neomycin resistance cassette is in intron 1 and the second LoxP site is inserted in intron 3. The FRT-flanked neomycin cassettes were removed in vivo by Flp-recombination by crossing to Flp-deleter breeding step. Hepatocyte specific FLAG-MTFP1 Knock-In (KI) mice (*Alb-Cre*[tg/+]*Mtfp1*[+/+], *CAG*[tg/+]) were generated by crossing an inducible mouse model for mCherry-P2A-FLAG-MTFP1[16] generated by PolyGene AG (Switzerland) on a C57Bl6/N background expression under the CAG promoter with mice expressing the Cre recombinase under the control of the hepatocyte-specific albumin Cre promoter. Liver specific *Mtfp1* knockout mice (LMKO) were generated by mating *Alb-Cre*[tg/+] mice with *Mtfp1*[LoxP/LoxP] mice, this led to the deletion of flanked region containing exons 2 and 3 (925 bp in total), resulting in the loss of most of the coding region of *Mtfp1* and a loss of *Mtfp1* expression. For fluorescence microscopy analysis of primary hepatocytes, the triple transgenic line *Alb-Cre*[tg/+] *mitoYFP*[tg/+]; *Mtfp1*[LoxP/LoxP] was generated. Mice were euthanized by cervical dislocation for all experiments, except for hepatocyte isolation, where mice under anesthesia were perfused through the inferior vena cava and the hepatic vein.

Mice were fed a normal chow diet (NCD, Teklad global protein diet; 20% protein, 75% carbohydrate, 5% fat) or high fat diet (HFD) containing 22 kJ% carbohydrates 24 kJ% protein and 54 kJ% fat (Ssniff® EF acc. E15742-347) for 16 weeks starting from the age of 8 weeks. During the treatment mice were monitored weekly for the body weight and every other week for the glycaemia, which was measured using a glucometer (Accu-Chek Performa, Roche, 1486023). To induce hepatic cell death, Fas antigen, CD95 (BD Biosciences, 554254) was intraperitoneal injected in 18-h fasted mice at the concentration of 225 μg/kg (BW). Mice were sacrificed 24 h post injection.

Mouse genotype was determined at weaning. For the genotyping, genomic DNA was isolated from mouse tail snip using lysis buffer Tris-HCl (EuroMedex, 26-128-3094-B) pH8.5 100 mM; Ethylenediaminetetraacetic acid (EDTA, 5 mM; EuroMedex, EU0084-B); Sodium Dodecyl Sulfate (SDS, 0.2%; EuroMedex, 1833); Sodium Chloride (NaCl, 200mM; EuroMedex, 1112-A); Proteinase K 400 μg/ml (Sigma-Aldrich, 03115879001); samples were incubated at 56 °C until digested and then centrifuged for 3 min at 16,000 g. Supernatants were transferred to fresh tubes, and DNA was precipitated first with 1000 μl of isopropanol, centrifuged at 16,000 g for 10 min at 4 °C, pellet were washed with 500 μl of 70% ethanol, centrifuged at 16,000 g for 10 min at 4 °C, and DNA was resuspended in $H_2O$. PCR was performed using

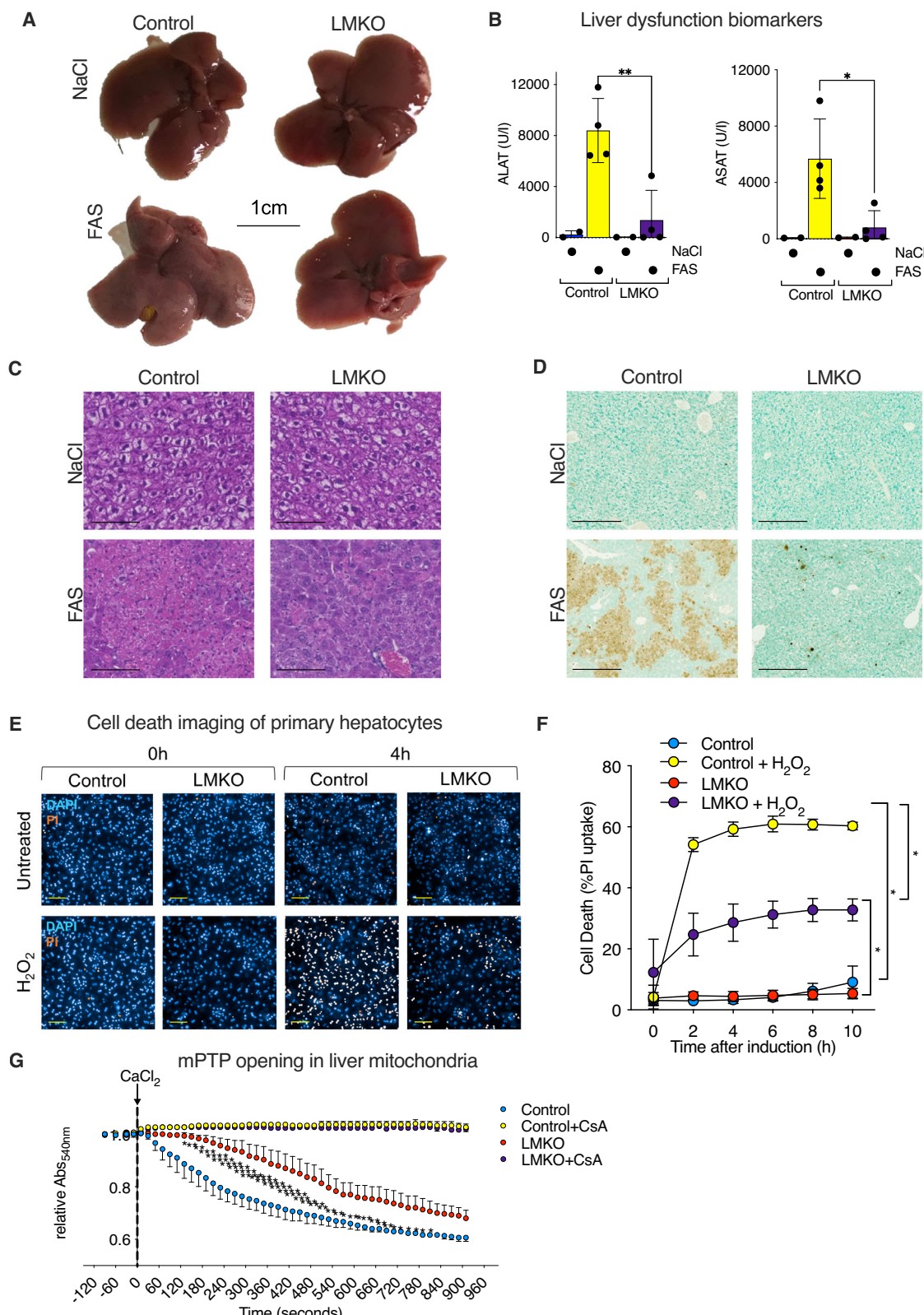

**Fig. 6 | Hepatic deletion of *Mtfp1* protects mice from FAS-induced liver damage. A** Representative images of livers from control ($n = 2$) and LMKO ($n = 4$) male mice treated with FAS or NaCl for 24 h. Scale bar = 1 cm. **B** Alanine (ALAT; left) and Aspartate transaminase (ASAT; right) levels in plasma of mice from **A**). Data are means ± SD. 2-tailed unpaired Student's *t* test, *$p < 0.05$, **$p < 0.01$. **C** H&E staining of control and LMKO livers from **A**) performed once for NaCl and four times for FAS with similar results. Scale bar = 100 μm. **D** TUNEL staining of control and LMKO livers from **A**) performed once. Scale bar = 100 μm. **E** Representative of high-content imaging of primary hepatocytes from control and LMKO male mice treated with or without $H_2O_2$ at 4 h [1mM] treatment. Blue: NucBlue (nuclei); Orange: propidium iodide (PI). Scale bar = 100 μm. **F** Cell death quantification of PI uptake of 3 independent experiments from **E**). Data are means ± SEM. 2-way ANOVA, *$p < 0.05$. **G** Mitochondrial permeability transition pore (mPTP) opening assay. Liver mitochondria from control and LMKO female mice were treated with $CaCl_2$ [120μM] to induce the mPTP opening by light scattering at 540 nm. Cyclosporin A (CsA) [1 μM] used to block mPTP opening. Data are means of 3 independent experiments ± SEM. 2-way ANOVA, *$p < 0.05$. **$p < 0.01$. ***$p < 0.001$. ****$p < 0.0001$.

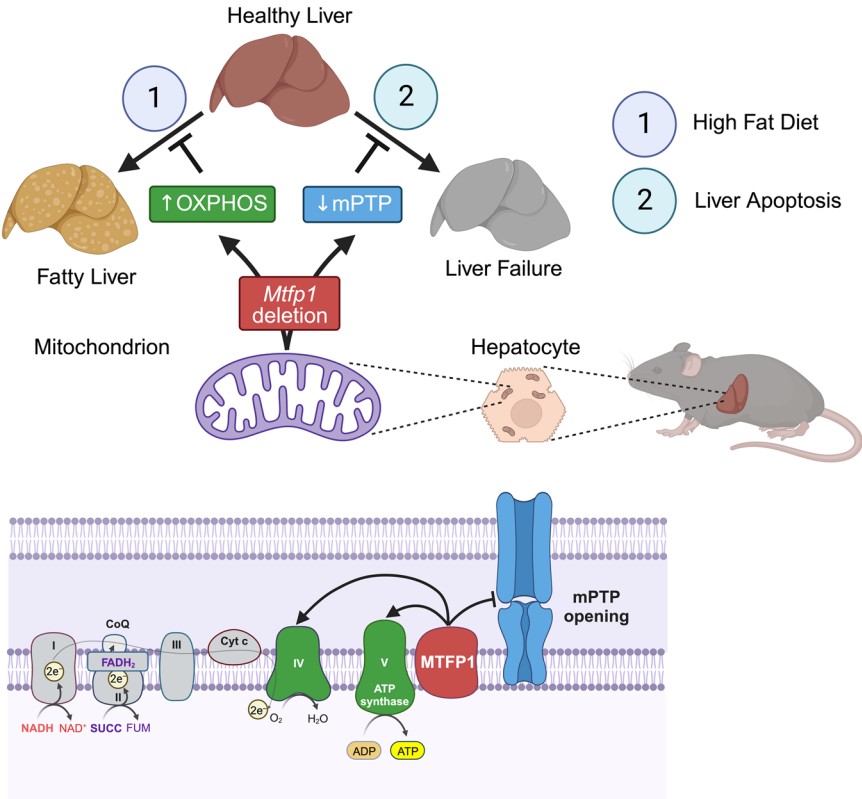

**Fig. 7 | Model for the regulation of liver resilience by MTFP1.** *Mtfp1* deletion in hepatocytes of liver-specific Mtfp1 knockout (LMKO) mice promotes liver resilience in vivo by 1) enhancing oxidative phosphorylation (OXPHOS) activity and resistance against fatty liver disease (hepatic steatosis) induced by chronic high fat diet (HFD) feeding and 2) increasing the resistance to mitochondrial permeability transition pore (mPTP) opening and liver apoptosis. Created with Biorender.com.

the 5x Hot FIREpol (SOLIS BIODYNE, 755015) at 94 °C for 3 min, (94 °C for 15 s, 55 °C 30 s, 72 °C for 1 min) for 35 cycles, 72 °C for 7 min and 4 °C. Primers used for genotyping are listed in Supplementary Dataset 5.

## Metabolic studies

Food and water intake, respiratory exchange ratio (RER), and physical activity were tested for 5 days using the PhenoMaster system (TSE Systems, Bad Homburg, Germany), which allowed continuous undisturbed and controlled recording. during day and night periods. 5 days before the recording, the animals were placed in the room and in cages similar to the PhenoMaster cages, for acclimatization. The following parameters were recorded every 3 min and averaged hourly or for the light and dark phase as indicated: food consumption (g), water consumption (ml), distance covered (cm), and RER (ratio between the amount of carbon dioxide ($CO_2$) produced in metabolism and oxygen ($O_2$) used). RER has a range from 0.7 to 1.0 according to the fuel source, when close to 0.7 means that fats are the predominant source, closest to 1.0, carbohydrates are predominantly consumed.

For Pyruvate tolerance test (IP-PTT): 18-h fasted mice were administered intra-peritoneal injections of sterile pyruvate (Sigma-Aldrich, P2256) at 1 g/kg (BW). Glucose and Insulin tests for Glucose tolerance test (IP-GTT): 18-h fasted mice were administered intra-peritoneal injections of sterile glucose (EuroMedex, UG3050) at 2 g/kg. For Insulin tolerance test (IP-ITT): 6-h fasted mice were administered intra-peritoneal injections of Insulin aspart (NovoRapid) at 1 g/kg (BW). For all the tests, blood glucose levels were measured before injection and 15, 30, and 60 post-injections using a glucometer (Accu-Chek Performa, Roche). For glucose-stimulated insulin production, mice were fasted for 18 h and were administered intra-peritoneal injections

of sterile glucose (EuroMedex, UG3050) at 2 g/kg. Insulin levels were analyzed by ELISA (Crystal Chem, 90082).

To assess the phosphorylation levels of key components in the insulin pathway, mice were subjected to a 5-h fasting period and subsequently injected with insulin (Humalog) at a dosage of 0.5 IU/kg. After a 15-min interval, the mice were euthanized, and their livers were promptly frozen using liquid nitrogen. The liver samples were then homogenized over ice in a lysis buffer containing 50 mM Tris (pH 7.4), 0.27M sucrose, 1mM EGTA, 1mM EDTA, 1% Triton X-100, 0.1% β-mercaptoethanol, and protease and phosphatase inhibitors. C2C12 lysates used in the blot were a kind gift from Eric Hajduch[80]. Whole blood was collected from the submandibular vein of awake mice in tubes containing heparin (Sigma-Aldrich, H4784) at the concentration of 50 U/ml and centrifuged at 3000 g for 10 min at 4 °C to isolate plasma. Plasma analyses of transaminases (ASAT and ALAT), LDH, cholesterol and triglycerides (TG) were performed by the "Microdosages biochimiques" at the "Institut de Physiologie et Pathologie, Claude Bernard" core facility (Paris).

## Analysis of body composition

Fat and lean mass were determined in awake Control and LMKO mice by nuclear magnetic resonance (NMR) using the Contactless Check Weighing (CCW) by MiniSpec MQ (Bruker MiniSpec Plus) with a 0.15-T magnetic field and a 6.2-MHz frequency pulse. This device measures the mass of fat, lean tissue, and fluid based on prior calibration. Mice were placed in an acrylic cylinder (48-mm diameter) and was loosely restrained within the cylinder by pushing a plunger. The cylinder was then positioned inside the bore of the magnet maintained at constant temperature of 37 °C. The measurements were recorded as % of fat, lean, and fluids.

## Tissue analyses

Histological and morphometric analysis: liver and white adipose tissue were fixed with phosphate-buffered 10% formalin (Sigma-Aldrich, F5554) for 20 h then washed in 70% ethanol, then embedded in paraffin, sectioned by microtome (4 um), and stained with hematoxylin & eosin. Cell death was assessed using the TUNEL assay kit – HRP – DAB (AbCam, ab206386). All slides were scanned using the Slide scanner Axio Scan.Z1 (ZEISS). Representative pictures were obtained using the CZI software (Zen 2.3 lite; ZEISS).

Triglyceride assay: snap frozen liver tissue was used to assess hepatic triglyceride content by fluorometric assay (AbCam, ab178780).

For transmission electron microscopy (TEM), small pieces ($1 \times 1 \times 1$ mm) from the livers of control and LMKO mice were fixed in a mixture of PHEM 1x buffer (60 mM PIPES, 25 mM HEPES, 10 mM EGTA, 2 mM $MgCl_2$; pH 7.3), 2.5% glutaraldehyde, and 2% PFA at 37 °C for 30 min, followed by an overnight fixation at 4 °C. Subsequently, tissues were rinsed three times with PHEM 2X buffer. Samples were then incubated with 1% osmium tetroxide (Merck) and 1.5% ferrocyanide (Sigma Aldrich) in 0.1M PHEM. After dehydration through a graded series of ethanol, samples were gradually infiltrated with epoxy resin at room temperature. Following heat polymerization, sections with a thickness of 70 nm were cut using a Leica UCT microtome and collected on carbon, formvar-coated copper grids. The sections were contrasted with 4% aqueous uranyl acetate and Reynold's lead citrate.

Ultra-large electron microscopy montages were generated at a magnification of 14500x (pixel size = 6.194 nm, bin 1, US4000 Ultrascan camera) using a TECNAI F20 Transmission Electron Microscope (FEI) with a field emission gun (FEG) as an electron source, operated at 200 kV. The SerialEM software[81,82] was employed for multiscale mapping, as previously described[24]. The 'Align Serial Sections/Blend Montages' interface of IMOD16 was utilized to blend the stacks of micrographs collected into single large images. Mitochondrial cross-sectional area quantification was determined by tracing the perimeter of individual mitochondria using 3dmod software

## Isolation of mitochondria

Livers were gently homogenized on ice in Mitochondria Isolation Buffer (MIB: Sucrose 275 mM (EuroMedex, 200-301-b); ethylene glycol tetra acetic acid (EGTA-KOH pH7 1mM; EuroMedex, E4378) Tris pH 7.4 20 mM; Bovine Serum Albumin (free fatty acid BSA, 0.25 mg/ml; Sigma-Aldrich, A6003) and protease inhibitors (Roche). Differential centrifugation at 1000 g for 10 min at 4 °C yielded supernatant that was subsequently centrifuged at 3000 g for 15 min at 4 °C. The resulting crude mitochondrial pellet fraction was resuspended in MIB and protein concentration was quantified with Bradford assay (Sigma-Aldrich, B6916) using a spectrophotometer (Infinite 200 Pro, Tecan). Isolated mitochondria were used for Blue-Native Polyacrylamide Gel Electrophoresis (BN-PAGE), SDS-PAGE or for High-Resolution Respirometry (HRR) assays.

## High resolution respirometry

Mitochondrial oxygen consumption was measured with an Oxygraph-2k (Oroboros Instruments, Innsbruck, AT). 50 to 500 μg of mitochondria were diluted in 2.1 ml of mitochondrial respiration buffer (Magnesium Chloride ($MgCl_2$, 3 mM; Sigma-Aldrich, M3634); Lactobionic Acid 60 mM (Sigma-Aldrich, 153516); Taurine 20 mM (Sigma-Aldrich, T0625); Potassium phosphate monobasic ($KH_2PO_4$, 10 mM; EuroMedex, 2018); 4-(2-Hydroxyethyl)piperazine-1-ethanesulfonic acid, N-(2-Hydroxyethyl)piperazine-N'-(2-ethanesulfonic acid) (HEPES-KOH pH 7.4 20 mM; EuroMedex, 10-110); Sucrose 110 mM; EGTA-KOH pH7 0.5 mM and free fatty acid BSA 1g/l). To ensure mitochondrial quality and integrity, cytochrome c (Cyt c, 2 μM;Sigma-Aldrich, C7752) was used. The oxygen consumption ($JO_2$) rate was measured using either Pyruvate 10 mM, Glutamate 5 mM (Sigma-Aldrich, G1626) and Malate 5 mM (Sigma-Aldrich, M1000), or Palmitoyl-Carnitine (PC,

25 uM; Sigma-Aldrich, P1645). To specifically assess Complex II activity, Rotenone 0.5 μM (Sigma-Aldrich, R8875) and Succinate 10 mM (Sigma-Aldrich, S2378) were used as substrates. Oxygen consumption was assessed in the phosphorylating state with Adenosine 5′-diphosphate monopotassium salt dihydrate (ADP, 0.2 mM; Sigma-Aldrich, A5285) or non-phosphorylating state by adding oligomycin (OLGM, 5 μM, Sigma-Aldrich, O4876).

The cytochrome c oxidase (COX) activity was measured in the presence of Ascorbate-Na (2 mM; Sigma-Aldrich, A4034; pH6 with Ascorbic acid (Sigma-Aldrich, A7506)), N,N,N′,N′-Tetramethyl-p-phenylenediamine dihydrochloride (TMPD, 0.5 mM; Sigma-Aldrich, 87890) and carbonyl cyanide m-chlorophenylhydrazone (CCCP, 0.2 μM, Sigma-Aldrich,C2759), with prior inhibition with Antimycin A (AMA, 2.5 μM; Sigma-Aldrich, A8674), finally KCN (2 mM, Sigma-Aldrich, 60178) was injected to evaluate the effective complex IV-dependent respiration.

The ATPase activity was assessed by colorimetric assay. 800 μg of liver mitochondria from LMKO and control mice were incubated for 15 min in incubation buffer (Triethanolamine 75 mM and $MgCl_2$ (2 mM pH = 8.9). Then, phosphate buffer (Molybdic acid 5.34 mM, $FeSO_4$ 28.77 mM and $H_2SO_4$ 393.96 mM) was added and incubated for 5 min and finally, absorbance was measured in the presence of ATP (5 mM). ADP (5 mM) and oligomycin (5 μM) were used as controls.

For the mitoplast respiration assay, permeabilized crude mitochondria were prepared using a freeze-thaw cycle[65]. Respiration rates were measured in a potassium phosphate buffer (50 mM, pH 7.2) with saturating concentrations of cytochrome c (62.5 μg/ml), NADH (0.625 mM), and succinate (10 mM). Oxygen consumption associated with complex I was determined following the addition of malonate, while rotenone was used to assess oxygen consumption linked to complex II.

## Mitochondrial swelling assay

Mitochondrial permeability transition pore opening was monitored by calcium-induced mitochondrial swelling and change of light scattering (absorbance 540 nm). Freshly isolated hepatic mitochondria from control and LMKO mice aged 8–10 week were resuspended in $Ca^{2+}$ uptake buffer (120 mM KCl, 5 mM MOPS, 5 mM $KH_2PO_4$, 10 mM Glutamate, 5 mM Malate, pH 7.4 in presence of cOmplete™ EDTA-free Protease Inhibitor Cocktail (Sigma 4693132001) at a concentration of 500 μg/mL and stimulated by the addition of a single pulse of 125 μM $CaCl_2$. The absorbance at 540 nm was measured at intervals of 30 s for 20 min at 37 °C using the microplate reader (Tecan). Cyclosporin A (1 μM) was used to inhibit the mPTP opening.

## BN-PAGE of liver mitochondria

1D BN-PAGE was performed as described previously[83] with some modifications. Briefly, liver mitochondria (100 μg, mitochondrial protein concentration was determined with DC Protein Assay BIO-RAD) were isolated from control and LMKO mice and incubated with digitonin extraction buffer (HEPES (30 mM), potassium Acetate (150 mM), Glycerol (12%), 6-Aminocaproic acid (2mM), EDTA (2 mM), high purity digitonin (4.5 g/g), pH = 7.2). Mitochondria were vortexed 1 h at 4 °C to solubilize membranes and after centrifuged at 21,000 g for 30 min. Supernatants were transferred to a new tube and mixed with loading dye (0.0124% (w/v), Coomassie brilliant blue G-250 (Invitrogen™, BN2004)). Digitonin solubilized mitochondria were loaded 3–12% Bis-Tris Invitrogen™ Novex™ NativePAGE™ 3–12% acrylamide gel (1 mm) (Invitrogen, BN2011BX10) using the anode running buffer (Invitrogen, BN2001) and Cathode Buffer Additive (0.5%) added to the anode buffer (Invitrogen, BN2002). Samples were migrated at 80V 20mA for 45 min and at 150V 20 mA for 13 h. Gel was incubated in Transfer buffer (Tris (0.304% (w/v), Glycine (1.44% w/v)) plus SDS (0.2% v/v) and β-mercaptoethanol (0.2% v/v)) for 30 min at room temperature (RT) to denature proteins. After the incubation, gels were transferred on

polyvinylidene difluoride (PVDF) in transfer buffer (Tris (0.304% w/v), Glycine (1.44% w/v), Ethanol (10%v/v)) at 400 mA 20V for 3 h and 30 min. After transfer, membranes were washed with methanol to remove Coomassie blue staining. For immunodetection, membranes were blocked with 5% milk in Tween-Tris-buffered saline (TTBS) for 1 h at RT and incubated overnight with the specific primary antibody diluted in blocking solution. The membrane was washed three times in TTBS and incubated for 2 h at RT with the appropriate secondary antibody conjugated with horseradish peroxidase (HRP). Finally, membranes were incubated in Tris-HCL (0.1 M pH 8.5) plus Luminol and p-Coumaric acid for 3 min and luminescence was detected using a ChemiDoc TM XRS+Imaging System. Band intensity was determined with Image Lab Software.

For 2D BN-PAGE, first dimension native gels were incubated in MOPS1x SDS Running Buffer (Fished Scientific) supplemented with β-mercaptoethanol (0.2% v/v) for 30 min at RT and then run in a denaturing second dimension electrophoresis (2D-SDS PAGE). The gels were run in MOPs buffer and then transferred to a nitrocellulose membrane. Immunodetection was performed as previously described. Primary antibodies used for immunoblotting of BN-PAGE are listed in Supplementary Dataset 5.

### Protein extraction, immunoblotting, and immunoprecipitation

To prepare total cell lysates for immunoblot analysis, cells were washed twice with cold Dulbecco's Phosphate-Buffered Saline (PBS; GIBCO, 14190169), scraped from the dishes and collected in tubes, then centrifuged at 16,000 g for 10 min at 4 °C. For liver samples, tissue was pulverized or isolated mitochondria were spun at 3000 g for 10 min at 4 °C. Cell or mitochondria pellet or tissue powder were resuspended in RIPA buffer (Tris pH 7.4 50 mM; NaCl 150 mM; Triton X-100 (EuroMedex, 2000C) 1%; Sodium Deoxycholate 0.05% (Sigma-Aldrich, D6750); SDS 0.10%; EDTA 1 mM; protease inhibitors and PhosStop Inhibitors (Roche), homogenized with a pestle (Eppendorf, 033522) and incubated at 4 °C for 30 min on a rotating wheel, then spun at 16,000 g for 10 min. Protein concentration was determined with Bradford assay using a plate reader (TECAN). Protein extracts were prepared in Laemmli Buffer (Bio-Rad, 1610747) with 2-Mercaptoethanol 355 mM (Sigma-Aldrich, M3148) and boiled at 95 °C for 5 min, then resolved by SDS-PAGE (Mini Protean chamber and TGX gels, Bio-Rad) and transferred onto nitrocellulose membrane (Trans-blot Turbo Transfer system and kit, Bio-Rad) followed by blocking with 3% milk for 30 min. Membranes were then incubated with the primary antibodies (listed in Supplementary Dataset 5) and HRP-linked secondary antibodies (anti-mouse or anti-rabbit IgG; Thermo Fisher Scientific, 10368172 and 11034). Clarity Western ECL (Bio-Rad) was used for the detection with ChemiDoc (Bio-Rad). Densitometric analysis of immunoblots was performed using Image Lab software (Bio-Rad). The signal of the quantified bands was normalized to loading controls or Stain-free TGX staining as indicated[84] and presented as fold difference over control treatment or control genotype as the invariant control.

### Proteomics

Liver extract proteins were extracted and denatured in RIPA buffer. Samples were sonicated using a Vibracell 75186 and a miniprobe 2 mm (Amp 80% // Pulse 10 off 0.8, 3 cycles) and further centrifuged. Protein assay was performed on supernatant (Pierce 660 nm, according to manufacturer instructions) and 100 μg of each extract was delipidated and cleaned using a Chloroform / Methanol/Water precipitation method. Briefly, 4 volumes of ice-cold methanol were added to the sample and vortex, 2 volumes of ice-cold chloroform were added and vortex and 3 volume of ice-cold water was added and vortex. Samples were centrifuged 5 min at 5000 g. The upper layer was removed and proteins at the interface were kept. Tubes were filled with ice-cold methanol and centrifuged at max speed for 5 min. Resulting protein

pellet was air-dried and then dissolved in 130 μl of 100 mM NaOH before adding 170 μl of Tris 50 mM pH 8.0, tris (2-carboxyethyl)phosphine (TCEP) 5 mM and chloroacetamide (CAA) 20 mM. The mixture was heated 5 min at 95 °C and then cooled on ice. Endoprotease LysC (1 μg) was use for an 8 h digestion step at (37 °C) followed with a trypsin digestion (1 μg) at 37 °C for 4 h. Digestion was stopped adding 0.1% final of trifluoroacetic acid (TFA). Resulting peptides were desalted using a C18 stage tips strategy (Elution at 80% Acetonitrile (ACN) on Empore C18 discs stacked in a P200 tips) and 30 μg of peptides were further fractionated in 4 fractions using poly(styrenedivinylbenzene) reverse phase sulfonate (SDB-RPS) stage-tips method as previously described[85,86]. Four serial elutions were applied as following: elution 1 (80 mM Ammonium formate (AmF), 20% (v/v) ACN, 0.5% (v/v) formic acid (FA)), elution 2 (110 mM AmF, 35% (v/v) ACN, 0.5% (v/v) FA), elution 3 (150 mM AmmF, 50% (v/v) ACN, 0.5% (v/v) FA) and elution 4 (80% (v/v) ACN, 5% (v/v) ammonium hydroxide). All fractions were dried and resuspended in 0.1% FA before injection.

LC-MS/MS analysis of digested peptides was performed on an Orbitrap Q Exactive Plus mass spectrometer (Thermo Fisher Scientific, Bremen) coupled to an EASY-nLC 1200 (Thermo Fisher Scientific). A home-made column was used for peptide separation ($C_{18}$ 50 cm capillary column picotip silica emitter tip (75 μm diameter filled with 1.9 μm Reprosil-Pur Basic $C_{18}$-HD resin, (Dr. Maisch GmbH, Ammerbuch-Entringen, Germany)). It was equilibrated and peptide were loaded in solvent A (0.1% FA) at 900 bars. Peptides were separated at 250 nl.min$^{-1}$. Peptides were eluted using a gradient of solvent B (ACN, 0.1% FA) from 3% to 7% in 8 min, 7% to 23% in 95 min, 23% to 45% in 45 min (total length of the chromatographic run was 170 min including high ACN level step and column regeneration). Mass spectra were acquired in data-dependent acquisition mode with the XCalibur 2.2 software (Thermo Fisher Scientific, Bremen) with automatic switching between MS and MS/MS scans using a top 12 method. MS spectra were acquired at a resolution of 35000 (at $m/z$ 400) with a target value of $3 \times 10^6$ ions. The scan range was limited from 300 to 1700 $m/z$. Peptide fragmentation was performed using higher-energy collision dissociation (HCD) with the energy set at 27 NCE. Intensity threshold for ions selection was set at $1 \times 10^6$ ions with charge exclusion of $z = 1$ and $z > 7$. The MS/MS spectra were acquired at a resolution of 17500 (at $m/z$ 400). Isolation window was set at 1.6 Th. Dynamic exclusion was employed within 45 s.

Data were searched using MaxQuant (version 1.5.3.8) using the Andromeda search engine[87] against a reference proteome of *Mus musculus* (53449 entries, downloaded from Uniprot the 24th of July 2018). The following search parameters were applied: carbamidomethylation of cysteines was set as a fixed modification, oxidation of methionine and protein N-terminal acetylation were set as variable modifications. The mass tolerances in MS and MS/MS were set to 5 ppm and 20 ppm, respectively. Maximum peptide charge was set to 7 and 5 amino acids were required as minimum peptide length. A false discovery rate of 1% was set up for both protein and peptide levels. All 4 fractions per sample were gathered and the iBAQ intensity was used to estimate the protein abundance within a sample[88].

The label-free quantitative (LFQ) proteomics data was annotated in Perseus v.1.6.14.0[89] with Mouse MitoCarta 3.0[26] for individual OXPHOS complexes, mitoribosome and whole mitochondria using uniprot IDs. The Relative Complex Abundance (RCA) of the OXPHOS and mitoribosome were plotted with an in-house R script (R v.4.0.3, R studio v.1.3.1093) correcting for differential mitochondrial content using the ratio mean of LFQ values of mitochondria "+" annotations from MitoCarta 3.0. The mean of normalized values and standard deviation were calculated for each subunit of each complex comparing the LMKO and control mouse values from quantitative proteomics data along with the 95% confidence interval based on the t-statistic for each complex. A paired $t$-test calculated the significance between the LMKO and control mouse values for each complex. $P$-value ***=$p \leq 0.001$, ns non-significant.

Mass spectrometry data have been deposited at the ProteomeXchange Consortium (http://www.proteomexchange.org) via the PRIDE partner repository[90,91] with the dataset identifier PXD041197.

## Co-Immunoprecipitation assay

500 µg of liver mitochondria were freshly isolated from liver tissue of hepatocytes specific Flag-MTFP1 Knock-In (KI) mice (*Alb-Cre^tg/+Mtfp1^+/+, CAG^tg/+*) and Control mice (*Albe-Cre^+/+Mtfp1^+/+, CAG^tg/+*) as described above. Mitochondria were lysed in IP buffer (20 mM HEPES-KOH pH 7.5, 150 mM NaCl, 0.25% Triton X-100, protease inhibitor cocktail) on ice for 20 min and then centrifugated at 10,000 g, 4 °C for 15 min. Supernatant obtained by centrifugation was then incubated with 20 µL of anti-FLAG magnetic beads (Sigma M8823) for 2 h at 4 °C. The immunocomplexes were then washed with IP buffer without Triton X-100 and eluted with Laemmli Sample Buffer 2x at 95 °C for 5 min. Protein were stacked in a 15% SDS-PAGE gel with a 10 min long migration at 80 V. Proteins were fixed in gel and migration was visualized using the Instant Blue stain (Expedeon). Bands were excised for digestion. Gel bands were washed twice in Ammonium bicarbonate (AmBi) 50 mM, once with AmBi 50 mM/ACN 50% and once with 100% ANC. Gel band were incubated for 30 min at 56 °C in 5 mM dithiothreitol (DTT) solution for reduction. Gel bands were washed in AmBi 50 mM and then in 100% ACN. Alkylation was performed at room temp in the dark by incubation of the gel bands in Iodocateamide 55 mM solution. Gel bands were washed twice in AmBi 50mM and in 100% ACN. 600 ng of trypsin were added for 8h digestion at 37 °C. Peptides were extracted by collecting 3 washes of the gel bands using AmBi 50 mM/50% ACN and 5% FA. Peptides clean up and desalting was done using Stage tips (2 disc Empore C18 discs stacked in a P200 tip).

LC-MS/SM analysis of digested peptides was performed on an Orbitrap Q Exactive HF mass spectrometer (Thermo Fisher Scientific, Bremen) coupled to an EASY-nLC 1200 (Thermo Fisher Scientific). A home-made column was used for peptide separation ($C_{18}$ 30 cm capillary column picotip silica emitter tip (75 µm diameter filled with 1.9 µm Reprosil-Pur Basic $C_{18}$-HD resin, (Dr. Maisch GmbH, Ammerbuch-Entringen, Germany)). It was equilibrated and peptide were loaded in solvent A (0.1% FA) at 900 bars. Peptides were separated at 250 nl.min$^{-1}$. Peptides were eluted using a gradient of solvent B (ACN, 0.1% FA) from 3% to 26% in 105 min, 26% to 48% in 20 min (total length of the chromatographic run was 145 min including high ACN level step and column regeneration). Mass spectra were acquired in data-dependent acquisition mode with the XCalibur 2.2 software (Thermo Fisher Scientific, Bremen) with automatic switching between MS and MS/MS scans using a top 12 method. MS spectra were acquired at a resolution of 60000 (at $m/z$ 400) with a target value of $3 \times 10^6$ ions. The scan range was limited from 400 to 1700 $m/z$. Peptide fragmentation was performed using HCD with the energy set at 26 NCE. Intensity threshold for ions selection was set at $1 \times 10^5$ ions with charge exclusion of $z = 1$ and $z > 7$. The MS/MS spectra were acquired at a resolution of 15,000 (at $m/z$ 400). Isolation window was set at 1.6 Th. Dynamic exclusion was employed within 30 s. Data are available via ProteomeXchange with identifier PXD046262.

Data were searched using MaxQuant (version 1.6.6.0) [1,2] using the Andromeda search engine [3] against a reference proteome of *Mus musculus* (53449 entries, downloaded from Uniprot the 24th of July 2018). A modified sequence of the protein MTP18 with a Flag tag in its N-ter part was also searched.

The following search parameters were applied: carbamidomethylation of cysteines was set as a fixed modification, oxidation of methionine and protein N-terminal acetylation were set as variable modifications. The mass tolerances in MS and MS/MS were set to 5 ppm and 20 ppm, respectively. Maximum peptide charge was set to 7 and 5 amino acids were required as minimum peptide length. A false discovery rate of 1% was set up for both protein and peptide levels. The iBAQ intensity was used to estimate the protein abundance within a sample.

Quantitative analysis was based on pairwise comparison of intensities. Values were log-transformed (log2). Reverse hits and potential contaminant were removed from the analysis. Proteins with at least 2 peptides (including one unique peptide) were kept for further statistics. Intensities values were normalized by median centering within conditions (normalizeD function of the R package DAPAR). Remaining proteins without any iBAQ value in one of both conditions have been considered as proteins quantitatively present in a condition and absent in the other. They have, therefore, been set aside and considered as differentially abundant proteins. Next, missing values were imputed using the impute.MLE function of the R package imp4. Statistical testing was conducted using a limma t-test thanks to the R package limma[92]. An adaptive Benjamini−Hochberg procedure was applied on the resulting *p*-values thanks to the function adjust.p of R package cp4p[93] using the robust method previously described[94] to estimate the proportion of true null hypotheses among the set of statistical tests. The proteins associated to an adjusted *p*-value inferior to a FDR level of 1% have been considered as significantly differentially abundant proteins. Mass spectrometry data have been deposited at the ProteomeXchange Consortium (http://www.proteomexchange.org) via the PRIDE partner repository[90,91] with dataset identifier PXD046262.

## Liver RNA sequencing and RT-qPCR

Total RNA was isolated from 50–100 mg of snap-frozen liver tissue by the NucleoSpin RNA kit (Macherey-Nagel, 740955). Quality control was performed on an Agilent BioAnalyzer. Libraries were built using a TruSeq Stranded mRNA library Preparation Kit (Illumina, USA) following the manufacturer's protocol. Two runs of RNA sequencing were performed for each library on an Illumina NextSeq 500 platform using paired-end 75bp. The RNA-seq analysis was performed with Sequana. In particular, we used the RNA-seq pipeline (version 0.9.13) (https://github.com/sequana/sequana_rnaseq) built on top of Snakemake 5.8.1[95]. Reads were trimmed from adapters using Cutadapt 2.10 then mapped to the mouse reference genome GRCm38 using STAR 2.7.3a[96]. FeatureCounts 2.0.0 was used to produce the count matrix, assigning reads to features using annotation from Ensembl GRCm38_92 with strand-specificity information[97]. Quality control statistics were summarized using MultiQC 1.8[98]. Statistical analysis on the count matrix was performed to identify differentially regulated genes, comparing different diets among same genotypes or different genotypes under same diet. Clustering of transcriptomic profiles were assessed using a Principal Component Analysis (PCA). Differential expression testing was conducted using DESeq2 library 1.24.0[99] scripts based on SARTools 1.7.0 indicating the significance (Benjamini−Hochberg adjusted *p*-values, false discovery rate FDR < 0.05) and the effect size (fold-change) for each comparison. Over-representation analysis (ORA) was performed to determine if genes modulated by HFD in control or LMKO mice are more present in specific pathways. ORA was performed on WebGestalt (https://www.webgestalt.org/). RNAseq data have been deposited at ENA with the dataset identifier E-MTAB-12920.

For RT-qPCR, 1 µg of total RNA was converted into cDNA using the iScript Reverse Transcription Supermix (Bio-Rad). RT-qPCR was performed using the CFX384 Touch Real-Time PCR Detection System (Bio-Rad) and SYBR® Green Master Mix (Bio-Rad) using the primers listed in Supplementary Dataset 5. *Gapdh* was amplified as internal standard. Data were analyzed according to the 2−ΔΔCT method[100].

## LC-MS metabolomics

Steady-state metabolomics experiments were carried out in collaboration with General Metabolics, LLC, and were executed at General Metabolics' laboratories in accordance with the methodology previously described[101].

## Primary hepatocyte isolation and imaging

Primary hepatocytes were isolated from 6–8 week-old mice as previously described[102]. Briefly, a catheter (22G feeding needle) was connected to a pump and inserted into the vena cava. Livers were perfused first with perfusion buffer (NaCl 0.15M; Potassium Chloride (KCl, 2.7mM; EuroMedex; P017); Sodium phosphate dibasic (Na$_2$HPO$_4$, 0.2mM; Sigma-Aldrich 255793); HEPES-KOH pH 7.4 10mM and EDTA 0.5 mM, pH 7.65) at the speed of 3.5 ml/min for 10 min, and then with the collagenase buffer (NaCl 0.15M, KCl 2.7mM, Na$_2$HPO$_4$ 0.2mM, HEPES-KOH pH 7.4 10 mM, Calcium Chloride (CaCl$_2$, 0.8 mM; Sigma-Aldrich C3881) and Collagenase type I 500 μg/ml (Thermo Fisher Scientific, 17018029). Liver lobes were collected in Washing Medium (William's medium (Thermo Fisher Scientific) supplemented with 10% Fetal Bovine Serum (FBS; GIBCO, 10270), a mix of penicillin (100 U/ml) and streptomycin (100 μg/ml) (Pen-Strep; Sigma-Aldrich, P4333) and amphotericin B (Fungizone, 250 ng/ml; GIBCO, 15290018)) and passed through a cell strainer. Cells were centrifuged at 40 g for 2 min, supernatant was removed and resuspended again with Washing Medium. This procedure was repeated 3 times. Finally, hepatocytes were counted with Trypan Blue using the Countless II FL Automated Cell counter (Invitrogen) and plated at 1.2 × 10$^5$ cells/cm$^2$ in Culturing Medium (William's medium supplemented with 20% FBS, a mix of Insulin (1.7 μM), Transferrin (68.75 nM) and Selenium (38.73 nM) (ITS; Sigma-Aldrich, I3146), dexamethasone (25 nM; Sigma-Aldrich, D4902), penicillin (100 U/ml) and streptomycin (100 μg/ml) and Fungizone (250 ng/ml) on pre-coated plates (collagen I (40 μg/ml, Gibco) and glacial acetic acid 0.1% (Sigma-Aldrich, A6283); and incubated at 37 °C then washed with PBS.

For cell death assays, primary hepatocytes were seeded in Cell Carrier Ultra 96-well plates (PerkinElmer, 6055302), treated with NucBlue Solution (Thermo Fisher)) (diluted 1 drop in 10 ml) for 30 min, then, after 2 washes in PBS, cells were treated or not with hydrogen peroxide (H$_2$O$_2$, 1 mM; Sigma-Aldrich, 95294) in the presence of Propidium Iodide (PI; Thermo Fisher, 640922) (1:500). Cells were immediately imaged with the Operetta CLS High-Content imaging system (PerkinElmer) at 37 °C and 5% CO$_2$. Images were taken every 2 h for 10 h with a 20x water-immersion objective (1.0 NA). PI and NucBlue were excited with the 530–560 nm and 355-385nm LEDs, respectively. Images were analyzed using Harmony 4.9 software (Perkin Elmer) using the analysis sequence in Supplementary Dataset 5.

Quantification of mitochondrial morphology: primary hepatocytes were isolated from mitoYFP+ mice and seeded in Cell Carrier Ultra 96-well plates, treated with 2 μM NucBlue for 30 min, then, after 2 washes in PBS. Cells were immediately imaged with the Operetta CLS High Content imaging system at 37 °C and 5% CO$_2$. Images were taken 16 h post-plating with a 63x water-immersion objective (1.15 NA). MitoYFP and Hoechst were excited with the 460–490 nm and 355–385 nm LEDs, respectively. Images were analyzed using the PhenoLOGIC supervised machine learning pipeline in Harmony 4.9 software (Perkin Elmer) with the analysis sequence in Supplementary Dataset 5. The algorithm was used to identify the best properties able to segregate into "Short" or "Long" populations according to their mitochondrial network. 50 cells of each population were manually selected to for supervised machine learning training. Dead cells identified by propidium iodide were excluded directly by the algorithm and not included in the quantification. To mimic HFD steatosis in primary hepatocytes isolated, culture medium of the cells was supplemented with Intralipid (0.65%) for 24 h. Intralipid is a complex lipid emulsion composed linoleic, oleic, palmitic, and stearic acids. BODIPY™ 558/568 C12 (Invitrogen D3835) was used to stain intracellular lipid accumulation.

To assess DRP1 colocalization with mitochondria in primary hepatocytes, cells were fixed with 100 μl of 4% paraformaldehyde per well at room temperature (RT) for 8 min.

After fixation, hepatocytes were washed three times with PBS for 5 min each. To permeabilize the cells, 0.5% (v/v) Triton/PBS was applied for 8 min (three times) on an orbital shaker at RT. The hepatocytes were then incubated with a blocking solution (5% [v/v] FBS/PBS) for 1 h on an orbital shaker at RT and immunostained with DRP1 antibody (26187-1-AP) and TOMM40 (18409-1-AP) : DRP1 (1:100) and TOMM40 (1:500) antibodies were diluted in the blocking solution and incubated overnight on an orbital shaker at 4 °C. The cells were washed three times with PBS for 5 min each, and a secondary antibody was incubated for 2 h on an orbital shaker at RT in darkness. After antibody incubation, the cells were washed with PBS and incubated with Hoechst (Invitrogen, H3570) diluted in PBS (1/2500) for 10 min on an orbital shaker at RT. Image acquisition was performed using the Operetta CLS High-Content Analysis system from Perkin Elmer, with a 63x objective. Subsequently, image analysis was conducted using the open-source software ImageJ/Fiji. For colocalization analysis, the Fiji plugin Coloc 2 was utilized. The images were preprocessed by applying a Gaussian blur filter, and the threshold was automatically set for all the images (default method).

## Hepatocyte cell line analyses

Huh-7.5 (RRID:CVCL_7927) (human) (a kind gift from Martin Kächele), Hep-G2 (RRID:CVCL_0027) (human) (HB-8065), HC-04 (RRID:CVCL_Z631) (human) (MRA-975), and Hepa 1–6 (RRID:CVCL_0327) (mouse) (CRL-1830) were used in the present study. For down-regulation of MTFP1, cells were transfected with either siGENOME Human *MTFP1* siRNA (Dharmacon) for human cell lines and Accell Mouse *Mtfp1* (67900) siRNA (Dharmacon) for the mouse cell line (20 nm) for a 12-h incubation period. As negative controls, cells with either siGENOME RISC-free siRNA (Dharmacon) for human cell lines or siGENOME Non-Targeting siRNA (Dharmacon) for the mouse cell lines. The siRNAs were preincubated with Lipofectamine RNAiMAX (Thermo Fisher Scientific, 13778150) in Opti-MEM (1X) (Gibco 31985-062) for 10 min prior to transfection. After 72 h from transfection, the cells were stained with NucBlue (1 drop in 10 mL) (Thermo Fisher Scientific, R37605) and Molecular Probes MitoTracker Deep Red FM (Thermo Fisher Scientific, M22426) (0.1 μM) for a 30-min incubation at 37 °C and 5% CO2.

Image acquisition was conducted using the Operetta CLS High-Content Analysis system from Perkin Elmer, with a 63x objective. Subsequently, image analysis was performed with the open-source software ImageJ/Fiji. For mitochondrial measurements, the total mitochondrial volume was calculated through trainable Weka segmentation (a Fiji plugin), followed by image binarization. MTFP1 depletion was assessed concomitantly by RT-qPCR using primers listed in Supplementary Dataset 5.

## Statistical analyses

Experiments were repeated at least three times and quantitative analyses were conducted blindly. Randomization of groups (e.g., different genotypes) was performed when simultaneous, parallel measurements were not performed (e.g., Oroboros, hepatocyte isolation). For high-throughput measurements (e.g., mitochondrial morphology, cell death), all groups were measured in parallel to reduce experimental bias. Statistical analyses were performed using GraphPad Prism v9 software. Data are presented as mean ± SD or SEM where indicated. The statistical tests used, and value of experiment replicates are described in the figure legends. Comparisons between two groups were performed by unpaired two-tailed $T$ test. To compare more than two groups or groups at multiple time points 1-way ANOVA or 2-way ANOVA was applied. Tests were considered significant at $p < 0.05$ (*$p < 0.05$; **$p < 0.01$; ***$p < 0.001$; ****$p < 0.0001$).

## Reporting summary

Further information on research design is available in the Nature Portfolio Reporting Summary linked to this article.

## Data availability

Source data for all experiments are available alongside this manuscript. The datasets generated in this study have been deposited in the in the European Nucleotide Archive (NEO) repository and Proteomics Identification Database (PRIDE). Accession numbers are as follows: RNAseq data (ENA: Project: E-MTAB-12920), liver proteome (PRIDE partner identifier PXD041197) and liver interactome (PRIDE partner identifier PXD046262). The processed RNAseq data, liver proteome and liver interactome are provided in Supplementary Datasets 1, 2, and 3. The complete datasets of fluorescence microscopy images generated and analyzed in the current study are not publicly available because exporting understandable file names linked to cell, treatment, and time identifiers from the Harmony 4.9, Perkin Elmer software is not feasible. However, these datasets are accessible from the corresponding author upon request. Source data are included with this paper. Source data are provided with this paper.

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

## Acknowledgements
We thank David Hardy and Maryse Moya-Nilges for the histology services, Priscilla Lopes and for technical assistance. We thank Nils-Göran Larsson for providing mitoYFP mice, Arnaud Mourier and Marie Lagouge for insightful discussions, and Marie Lemesle for excellent administrative assistance. T.W. is supported by the European Research Council (ERC) Starting Grant No. 714472 (Acronym "*Mitomorphosis*") and the Agence Nationale pour la Recherche (ANR-20-CE14-0039-02). D.A.S. and D.H.H. are supported by grants from the Australian National Health and Medical Research Council (GNT2009732) and Medical Research Future Fund (MRF2016030). We acknowledge the Bio21 Mass Spectrometry and Proteomics Facility (MMSPF) for the provision of instrumentation, training, and technical support.

## Author contributions
T.W., C.P., J.D.H.C and E.D. designed the experiments. C.P. performed the in vivo experiments. C.P., J.D.H.C., E.V. and S.Y. performed the in vitro experiments. T.Co performed the RNAseq analyses. T.Ch performed the proteomics experiments and Q.G.G. performed the LFQ analyses. D.H. performed the RCA proteomics analyses, supervised by D.S.; M.M. is the supervisor of the proteomics platform. A.G. performed the TEM analyses and IN supervised the metabolic cage studies. C.P., J.D.H.C., E.D. and T.W. participated in the data interpretation, and manuscript writing. TW performed study supervision and obtained funding for the study.

## Competing interests
The authors declare no competing interests.
