## [Peer Review File · Nature Communications]

Mtfp1 ablation enhances mitochondrial respiration and protects against hepatic steatosisREVIEWER COMMENTS

Reviewer #1 (Remarks to the Author):

Overall, this manuscript provides valuable insights into the role of hepatic MTFP1 (mitochondrial fission process 1), an inner membrane protein, in regulating mitochondrial OXPHOS activity and mPTP opening and its impact on obesity and metabolic disorders. The authors discovered that hepatic MTFP1 controls mitochondrial OXPHOS activity and respiration, while not significantly impacting mitochondrial dynamics. Through comprehensive transcriptomics and proteomics analyses, the study revealed that deletion of MTFP1 in the liver leads to increased levels of complex V components, such as ATP5B. Notably, the authors identified relevant binding partners of MTFP1 through interactome analysis, including proteins involved in mitochondrial translation and OXPHOS, such as SLC25A family proteins. Additionally, the authors provided compelling evidence that hepatic MTFP1 deletion protects mice from high-fat diet-induced obesity and metabolic disorders. To thoroughly characterize the phenotypes associated with hepatic MTFP1 deletion, which exhibit intriguing differences from those observed in heart-specific MTFP1 deletion, the authors employed a combination of transcriptomics, proteomics, and interactomics analyses in primary hepatocytes and livers of control and liver-specific MTFP1 knockout mice. The manuscript is well-written, and the figures are well-organized. However, it is important to note that the molecular mechanisms underlying MTFP1's actions remain unexplored. Several omics data presented in the manuscript lack clear connections or linkages to explore the MTFP1 function. The direct link between the functions/roles of MTFP1 and the observed phenotypes needs to be examined. Furthermore, the MTFP1 role in the regulation of OXPHOS and mPTP was investigated already in the previous paper (Donnarumma et al, 2022, Nat Commun). The detailed concerns that should be addressed to enhance the clarity and impact of the findings are listed below:

Major concerns:

1. Regarding the role of hepatic MTFP1 in mitochondrial dynamics, Figure S2C shows that the deletion of hepatic MTFP1 slightly elongates mitochondria in primary hepatocytes, which contradicts the TEM data (Figure S2E). To determine if hepatic MTFP1 contributes to

mitochondrial fission, it would be helpful to quantify mitochondrial length in the TEM data and investigate the localization of DRP1 and other fission factors in MTFP1-deleted primary hepatocytes.

2. OPA1 and other proteins relevant to mito dynamics were identified as MTFP1 interacting proteins (Figure 3D). Does MTFP1 deletion alter their localization, modifications, and expression levels?

3. Since SLC25A family proteins were identified as direct interacting partners of MTFP1 by Co-IP proteomics, it would be interesting to explore if SLC25A activity is affected by MTFP1 deletion and if the MTFP1-SLC25A interaction contributes to the regulation of levels of proteins identified by proteomics in Figure 2, such as ATP5B and UQCRC2.

4. To further clarify the function of MTFP1, it would be valuable to investigate if MTFP1 directly binds to SLC25A family proteins and controls their activity.

5. The authors have identified ATP5B and other OXPHOS proteins as downstream targets of MTFP1 through proteomics. To further elucidate their roles in increased fat oxidation and respiration and the reduced sensitivity of mPTP observed in MTFP1 KO primary hepatocytes or livers, it would be valuable to investigate the contribution of these proteins by depleting or deleting them in MTFP1 KO cells or tissues and examine if the downregulation of ATP5B and other proteins reverses the phenotypes observed in MTFP1 KO.

6. Lipotoxicity in the liver is widely recognized as a contributor to insulin resistance. The observation that the LMKO liver exhibits reduced lipid accumulation and lipoatrophy compared to control livers suggests a potential improvement in insulin resistance induced by a high-fat diet (HFD) in LMKO mice. However, it is noteworthy that no significant difference in insulin sensitivity was observed between the control and LMKO mice. To gain deeper insights into the insulin sensitivity of liver tissues, it would be beneficial to investigate the insulin signaling pathway, including the phosphorylation status of key components such as IRS, IR, and AKT. By examining the phosphorylation levels of these proteins, a more comprehensive understanding of the impact of hepatic MTFP1 deletion on

insulin signaling and sensitivity in the liver can be obtained.

7. It would be interesting to explore how the liver-specific MTFP1 KO affects white adipose tissue (WAT) and brown adipose tissue (BAT) to gain a more comprehensive understanding of the systemic effects of hepatic MTFP1 deletion.

8. The manuscript does not show a difference in respiratory exchange ratio (RER) between the dark and light cycles, despite the expectation of increased RER during the dark cycle due to more food intake. It would be valuable to discuss this discrepancy and provide potential explanations for the observed results.

Minor comments:

1. The manuscript mentions that MTFP1 deletion in primary hepatocytes increases the activity of Complex IV and V, as well as fatty-acid-derived energy metabolism. However, the proteomics data only identified ATP5B and not other proteins such as NDUFA9, SDHA, and MTCO2 (Figure 2I and J). It would be helpful to explain why Complex IV and fatty-acid-derived energy metabolism were increased by MTFP1 deletion without affecting the levels of these proteins.

2. In Figures 2E and 2F, MTFP1 deletion led to a 20% increase in the activities of Complex IV and V. It would be valuable to explain and discuss how this increase translates to a 200% increase in respiration, as shown in Figure 2B.

3. While the omics data presented in the manuscript are insightful, there is a need for better integration and linkage of these data to provide a more cohesive and comprehensive understanding of the role of MTFP1 in hepatic mitochondria. It would be beneficial to discuss the relationships between the different omics findings and how they collectively contribute to the overall conclusions of the study.

4. In Figure 3C, no bands for MTFP1-binding proteins were observed. It would be advantageous to improve the figure quality to ensure clear visualization of the protein

bands and their interactions with MTFP1.

Reviewer #2 (Remarks to the Author):

I read with attention the paper by C. Patitucci et al. and I enjoyed the highly instructive content enriched with several explanatory experiments. The paper provides evidence of the importance of mitochondrial function in liver steatosis and the concept that mitochondrial function must be preserved in conditions accumulating intrahepatic fat.

In particular, the authors identify Mitochondrial Fission Process 1 protein (MTFP1) as a key regulator of mitochondrial and metabolic activity in the liver. They show that deletion of Mtfp1 in hepatocytes is physiologically benign in mice but leads to the upregulation of oxidative phosphorylation (OXPHOS) activity and mitochondrial respiration. This effect is, independent of mitochondrial biogenesis. Hepatocyte-specific knockout mice are protected against high fat diet-induced hepatic steatosis and metabolic dysregulation. The deletion of Mtfp1 in liver mitochondria inhibits mitochondrial permeability transition pore opening in hepatocytes, conferring protection against apoptotic liver damage in vivo and ex vivo. The function of MTFP1 in the liver points to its regulatory function of OXPHOS.

The methodology is well described and rather complete, and results explained satisfactorily within each paragraph.

The findings are novel, although some translational concepts will increase the impact of this paper.

I provide some comments which will help the authors to strengthen their work.

General

The discussion requires few lines of comments about the potential translational value of the major findings. At the end of the abstract the authors conclude that “...MTFP1.....positioning this gene as an unexpected regulator of OXPHOS and a therapeutic candidate for NAFLD. The same comment needs to be extended at the end of the discussion.

Abstract

-the text is unbalanced with too much background. Please describe more methods/results.

Introduction

- The authors mention the impact of NAFLD but they should also mention the debate about new definitions like MAFLD or even more recently MASLD. This is important in the overall classification of liver steatosis without using “stigmatising” words like “alcoholic” and “fatty”. Please consider these papers doi 10.1097/hep.0000000000000520 PMID 01515467-990000000-00488;

doi 10.1007/s11739-023-03203-0 PMID 36807050. After that, please decide what will be the acronym adopted throughout the text, i.e., NAFLD?, MAFLD?, MASLD? Etc.

- Please refer to the percent of subgroups having the chance to progress from simple steatosis to steatohepatitis to cirrhosis and, eventually hepatocellular carcinoma.

Results/Methods

- Please clarify the difference between “liver-specific MTFP1 KO” and “hepatocyte-specific MTFP1 KO”. Otherwise use same terminology throughout the text

Discussion

- Line 344 parenthesis is missing

Figure 6B

- Drawing breaks on the Y axis will increase the visibility of the lowest bars

Figure 7

The figure is nicely depicted, although I suggest being more detailed about mechanisms and pathways. This would be more in line with the amount of experimental data accumulated in the paper.

We thank the two reviewers for their critical appraisal of our manuscript and for their constructive criticism and comments. We have addressed each comment and question raised by the reviewers below in point-by-point responses.

REVIEWER COMMENTS

Reviewer #1 (Remarks to the Author):

Overall, this manuscript provides valuable insights into the role of hepatic MTFP1 (mitochondrial fission process 1), an inner membrane protein, in regulating mitochondrial OXPHOS activity and mPTP opening and its impact on obesity and metabolic disorders. The authors discovered that hepatic MTFP1 controls mitochondrial OXPHOS activity and respiration, while not significantly impacting mitochondrial dynamics. Through comprehensive transcriptomics and proteomics analyses, the study revealed that deletion of MTFP1 in the liver leads to increased levels of complex V components, such as ATP5B. Notably, the authors identified relevant binding partners of MTFP1 through interactome analysis, including proteins involved in mitochondrial translation and OXPHOS, such as SLC25A family proteins. Additionally, the authors provided compelling evidence that hepatic MTFP1 deletion protects mice from high-fat diet-induced obesity and metabolic disorders. To thoroughly characterize the phenotypes associated with hepatic MTFP1 deletion, which exhibit intriguing differences from those observed in heart-specific MTFP1 deletion, the authors employed a combination of transcriptomics, proteomics, and interactomics analyses in primary hepatocytes and livers of control and liver-specific MTFP1 knockout mice.

The manuscript is well-written, and the figures are well-organized. However, it is important to note that the molecular mechanisms underlying MTFP1's actions remain unexplored. Several omics data presented in the manuscript lack clear connections or linkages to explore the MTFP1 function. The direct link between the functions/roles of MTFP1 and the observed phenotypes needs to be examined. Furthermore, the MTFP1 role in the regulation of OXPHOS and mPTP was investigated already in the previous paper (Donnarumma et al, 2022, Nat Commun). The detailed concerns that should be addressed to enhance the clarity and impact of the findings are listed below:

Major concerns:

1. Regarding the role of hepatic MTFP1 in mitochondrial dynamics, Figure S2C shows that the deletion of hepatic MTFP1 slightly elongates mitochondria in primary hepatocytes, which contradicts the TEM data (Figure S2E). To determine if hepatic MTFP1 contributes to mitochondrial fission, it would be helpful to quantify mitochondrial length in the TEM data and investigate the localization of DRP1 and other fission factors in MTFP1-deleted primary hepatocytes.

We thank the reviewer for this helpful suggestion. As suggested, we have now quantified mitochondrial length in our original TEM data and observe no significant differences between control and LMKO liver mitochondria (Fig. S2E, right graph). These data are consistent with and corroborate the previous quantification of surface area originally reported in Fig. S2E (left graph). We did indeed report a slight elongation of mitochondrial length in isolated primary hepatocytes using a supervised machine learning-assisted quantification workflow that was also applied in a similar fashion to MTFP1 knockout fibroblasts in our previous study¹. However, this is not a direct measurement of mitochondrial length nor area but rather a single-cell based classification approach that forces individual cells to be attributed to user-defined categories of “elongated” or “tubular”, which may account for this discrepancy along with the fact that in situ measurements of mitochondrial morphology by TEM (Fig. S2E) may differ from ex vivo measurements of mitochondrial morphology in primary hepatocytes (Fig. S2A). Finally, we performed new experiments in which we measured mitochondrial morphology in

four different hepatocyte cell lines depleted for MTFP1 (Fig. S3A-D). We observed no mitochondrial elongation as assessed quantified surface area, perimeter, circularity or diameter. These data indicate that MTFP1 is not required for mitochondrial fission and are consistent with previous studies that have demonstrated that mitochondrial fission can be induced independently of MTFP1 expression in the liver^{2,3}.

We have also now assessed the cellular localization of DRP1 in primary hepatocytes and proximity to mitochondria by new immunofluorescence experiments in control and LMKO primary hepatocytes (Fig. S2G). We observed no significant differences in the localization of DRP1 with mitochondria between control and LMKO primary hepatocytes, consistent with previous reports in mouse fibroblast¹ and our observation that MTFP1 is dispensable for mitochondrial fission. Finally, we assessed the localization and steady-state levels of DRP1, as well as other mitochondrial dynamics proteins including FIS1, MFF, and OPA1 by fractionation and immunoblotting of control and LMKO mouse livers (Fig. S2H, see Fig. below) and observed no differences in their recruitment, localization and expression. Moreover, we observed to differences in OPA1 processing by YME1L and OMA1 as evidenced by unaltered ratios of L-OPA1 to S-OPA1, indicating the OPA1 proteolysis is not altered by MTFP1 ablation (See figure below).

2. OPA1 and other proteins relevant to mito dynamics were identified as MTFP1 interacting proteins (Figure 3D). Does MTFP1 deletion alter their localization, modifications, and expression levels?

This is a very salient point that we previously explored in MTFP1-deficient fibroblasts and tissues¹. Now, we have experimentally addressed this question in the LMKO mice in this study with new experiments. OPA1 is a nuclear-encoded protein that is imported into mitochondria, processed by the mitochondrial processing peptidase (MPP) to remove the N-terminal mitochondrial targeting sequence, yielding long OPA1 (L-OPA1) isoform embedded in the inner mitochondrial membrane (IMM) that is subsequently cleaved by the mitochondrial proteases YME1L and OMA1 to generated soluble, short OPA1 (S-OPA1) lacking the membrane anchor⁴. New immunoblot analyses (see figure from point 1 above) show total OPA1 levels to be unaffected in LMKO mitochondria. These data corroborate the original Omics analyses performed on LMKO livers that also revealed no differences in OPA1 protein (Fig. S2F, Supplemental Dataset 2) nor *Opa1* mRNA levels (Supplemental Dataset 1) nor any other proteins that directly control mitochondrial fission or fusion (Fig. S2F). Moreover, we did not observe any OPA1 bands above the expected molecular weight of L-OPA1, indicating that MPP-dependent processing of the precursor OPA1 occurs normally in the matrix (see figure above). This means that in the absence of MTFP1 in the liver mitochondrial import of OPA1 is normal, consistent with what we and others have reported in other tissues and cell lines^{1,5}. Finally, quantification of S-OPA1/L-OPA1 revealed no differences in LMKO mitochondria (see figure above), demonstrating that as in mouse fibroblasts and cardiomyocytes¹ and kidney⁵, MTFP1 depletion in the liver does not incite aberrant proteolytic processing of OPA1. Thus,

there is no evidence to suggest that OPA1 is modulated by MTFP1. Complexome profiling studies performed in heart mitochondria show no overlap of OPA1-containing and MTFP1-containing complexes (see figure below), and this is also evident from new immunoblot analyses of 2D-BN-PAGEs (see figure below) indicating that the physical interaction between OPA1 and MTFP1 is likely indirect and could reflect the crowded environment of the inner mitochondrial membrane since OPA1 complexes and MTFP1 complexes migrate at different molecular weights.

MTFP1 baits beyond OPA1 that we identified in Co-IP experiments that are involved directly or indirectly in mitochondrial dynamics are MFN2, ATAD3A, CHCHD2 and CHCHD3 (MIC19). None of these proteins showed altered levels of steady state protein (Supplemental Dataset 2) or mRNA (Supplemental Dataset 1) expression. New immunoblot analyses (see figure below) did not reveal aberrant migration of MFN2, which is one of the two mitofusins required for OMM fusion. CHCHD2 and CHCHD3 are proteins that are imported into the IMS in a redox relay pathway dependent on MIA40 (CHCHD4). We observed no differences in CHCHD2 or CHCHD3 levels, suggesting that biogenesis of these proteins is unaffected. ATAD3A is a pleiotropic protein that has previously been linked to mitochondrial dynamics as well as cholesterol biosynthesis, mtDNA maintenance, and innate immune signaling⁶⁻⁹. Its precise impact on mitochondrial dynamics notwithstanding, we do not observe differences in steady state levels.

To better highlight the impact (or lack thereof) that MTFP1 ablation has on mitochondrial dynamics proteins, we have now included volcano plots specifically representing these proteins from the LFQ proteomics of control and LMKO liver lysates (Supplemental Dataset 2) and call out this new graph within the text (Fig. S2F). From these data, we can observe that none of the fission and fusion-related genes are dysregulated in MTFP1-deficient livers, except for SLC25A46. Substantial depletion of SLC25A46 would be predicted to cause mitochondrial fragmentation¹⁰, which we do not observe in cell lines (Fig. S-D), primary hepatocytes (Fig. S2C), or in situ liver sections (Fig. S2E), indicating that the 50% reduction we detect by proteomics ($\text{Log}_2\text{FC} = 1.00409284$) is insufficient to elicit a functional response. Taken together, our data do not support an effect on mitochondrial dynamics by MTFP1 in the liver.

3. Since SLC25A family proteins were identified as direct interacting partners of MTFP1 by Co-IP proteomics, it would be interesting to explore if SLC25A activity is affected by MTFP1 deletion and if the MTFP1-SLC25A interaction contributes to the regulation of levels of proteins identified by proteomics in Figure 2, such as ATP5B and UQCRC2.

The MTFP1 liver interactome (Fig. 3D) study did indeed reveal SLC25A22, SLC25A11, SLC25A20 and SLC25A15 to co-immunoprecipitated with MTFP1. However, we cannot conclude that any of these proteins **directly** interact with MTFP1, as stated by the reviewer. As for OPA1, we surmise that these interactions to be indirect. While no published complexome profiling of A11, A22, nor A15 exist, the migration of A20 does not co-migrate with MTFP1, suggesting that they exist in a macromolecular complex that is distinct from that of MTFP1.

To directly assess the impact of MTFP1 on the carrier activity of these SLC25A proteins would require co-expression of MTFP1 and reconstitution in liposomes in which radiolabeled uptake of SLC25A substrates could be tracked, which has been the gold standard established by the Palmieri group for SLC25A family members. As we lack the technical expertise for in vitro reconstitution as well as permits and infrastructure to carry out radioactive studies, we decided to perform new metabolomic studies on the control and LMKO liver biopsies coming from the same mice used for proteomics and transcriptomic studies (Fig. S4A-D). These data have now been included in the revised manuscript (Fig. S4A-D, Supplemental Dataset 4 - Metabolomics) and are summarized alongside a description of the function of the candidate SLC carriers below.

SLC25A22 is a glutamate carrier that is highly expressed in the liver, as well as the pancreas and spleen¹¹. Glutamate, which is produced from oxoglutarate and ammonium by glutamate dehydrogenase, is exported out of the mitochondria by SLC25A22. The new metabolomics

data reveal no significant differences in the levels of glutamate, 2-oxoglutarate, nor oxoglutarate between control and LMKO fed a normal chow diet.

SLC25A11 is an oxoglutarate carrier that mediates the export of oxoglutarate in exchange for a dicarboxylate (usually malate) and is an important component of the malate-aspartate shuttle. The malate oxoglutarate carrier (OGC) imports malate, which undergoes conversion to oxaloacetate (OAA). Subsequently, OAA is transformed back into oxoglutarate through the action of glutamate OAA transaminase, which also concurrently converts glutamate to aspartate. Finally, the resulting oxoglutarate is exported by the OGC. We observed no differences in the steady state levels of either malate or aspartate. Moreover, malate-dependent state 2 JO_2 rates measured in isolated mitochondria from LMKO mice (in the absence of exogenous ADP) was already shown to be normal (Fig2B), consistent with the notion that SLC25A11 activity is unaffected by MTFP1 ablation in the liver.

SLC25A20 functions as a transporter for carnitine and acylcarnitine. In rodents, this carrier, known as CAC, exhibits a similar role, displaying a stronger preference for medium and long-chain carnitine esters compared to short-chain carnitines and L-carnitine. CAC expression is notably high in the liver and cardiac muscle. This transporter plays a crucial role in β -oxidation. In this process, cytosolic fatty acids are transferred to carnitine by CPT1 (carnitine palmitoyltransferase-1) situated in the outer membrane. Subsequently, the resulting acylcarnitine is transported into the matrix by CAC in exchange for carnitine. Metabolomic profiling of acylcarnitine species showed no major differences in the acylcarnitine metabolites examined: L-Acetylcarnitine, 2-Methylbutyrylcarnitine, 2-Hexenoylcarnitine, Hexanoylcarnitine, hydroxyisovaleryl_carnitine, Heptanoylcarnitine, Glutaryl_carnitine, Hydroxyhexanoylcarnitine, 3-hydroxyhexanoyl, 3-Hydroxy-cis-5-tetradecenoylcarnitine, Pentadecanoylcarnitine. We observed an upregulation in only two acylcarnitine species out of the eleven that were examined. (2-Methylbutyrylcarnitine and Hexanoylcarnitine). These results do not support a role of MTFP1 ablation in the liver with the modulation of SLC25A20 activity.

The gene *SLC25A15* encodes two ornithine carriers, namely ORNT1 and ORNT2, both of which are found in the liver and are essential components of the urea cycle. ORNT1, in addition to its role in the urea cycle, has the capacity to transport the amino acids lysine and arginine. On the other hand, ORNT2 is capable of transporting histidine. Mutations in the *SLC25A15* gene, specifically responsible for encoding ORNT1, lead to a neurological disease characterized by hyperornithinaemia, hyperammonaemia, and homocitrullinuria¹². If SLC25A15 activity were impacted by MTFP1 ablation, we would expect altered levels of these metabolites. However, metabolomic profiling revealed unaltered levels of ornithine, acetylorntithine, succinyl-l-ornithine, lysine, and arginine, suggesting that this is unlikely to be the case.

As mentioned previously, the interactions between MTFP1 and SLC25 carriers are unlikely to be responsible for the increased levels of UQCRC2 (complex III subunit) and ATP5B (complex V subunit). Steady-state levels of SLC25A22, 11, and 20 proteins are not altered in LMKO mice and individual depletion of each of these carriers has not been shown to phenocopy MTFP1 ablation nor lead to increased OXPHOS activity¹³⁻¹⁶. Compelling evidence implicates SLC25A29, SLC25A47, and SC25A39 in the regulation of OXPHOS activity¹⁷⁻¹⁹, yet none of these were found to be altered in LFQ proteomics or interactomic studies of liver MTFP1 mice. Hence, we cannot produce evidence to support the hypothesis that alteration in SLC25A activities of these 4 specific carriers could account for the observed differences in Complex IV and V activities in LMKO liver mitochondria.

4. To further clarify the function of MTFP1, it would be valuable to investigate if MTFP1 directly binds to SLC25A family proteins and controls their activity.

If the binding was direct, we would predict that SLC25A family proteins would co-migrate in a complex with MTFP1. This appears to be the case for SLC25A4 (ANT1), but the activity of ANT1 does not seem to be affected based on our bioenergetic measurements and membrane potential measurements. We have shown by 2D-BN-PAGE that the MTFP1-containing complexes migrates similarly in liver (this study) as they do in heart¹. As mentioned previously, the migration of native complexes containing SLC25A20 measured by complexome profiling studies in rodent heart²⁰ are incongruent with MTFP1 complexes. Therefore, we do not have any biochemical or metabolomic evidence to suggest that SLC25A interactors of MTFP1 are functionally impacted.

5. The authors have identified ATP5B and other OXPHOS proteins as downstream targets of MTFP1 through proteomics. To further elucidate their roles in increased fat oxidation and respiration and the reduced sensitivity of mPTP observed in MTFP1 KO primary hepatocytes or livers, it would be valuable to investigate the contribution of these proteins by depleting or deleting them in MTFP1 KO cells or tissues and examine if the downregulation of ATP5B and other proteins reverses the phenotypes observed in MTFP1 KO.

We fully agree with the reviewer about the value in determining whether Complex V increase is directly responsible for the increased mitochondrial respiration and reduced lipid accumulation. The increased JO_2 rates measured we reported in the manuscript when provided either with pyruvate, glutamate and malate (PGM), succinate/rotenone, and malate/PC under state 3 conditions (Fig. 2B) without a commensurate alteration in mitochondrial membrane potential (Fig. 2D) can therefore be attributed to a concomitant increase in both Complex IV and V activities, which we showed in Fig. 2E and F.

Genetic ablation of ATP5B in LMKO mice is unlikely to be informative since genetic deletion of this structural subunit (**Atp5f1b**^{tm1b(EUCOMM)Hmgw}) is perinatally lethal with complete penetrance according to the IMPC in C57Bl6/Ncr1 mice (which is the genetic background used in our study). Conditional *Atp5b*^{LoxP/LoxP} mice are commercially available but only on the C57Bl6/J background, which would involve 2-3 years of backcrossing with LMKO and control mice before liver-specific ablation of ATP5B in control and LMKO mice could be conducted and thus is beyond the scope of this study. Therefore, we elected to test this hypothesis in cellulo by depleting MTFP1 in mouse and human hepatocyte cell lines and subjecting these cells to phenotypic characterizations performed in vivo and ex vivo. We were able to achieve efficient depletion of MTFP1 in human (Huh7.5, HepG2 and HC04) and mouse (Hepa1.6) hepatocyte cells and this did not lead to increase ATP5B levels. Commensurately, mitochondrial respiration was also unaffected by MTFP1 depletion (see figure below). This is consistent with the notion that increased hepatic steatosis in MTFP1-deficient liver requires an upregulation of Complex V. The discrepancy between our observations in hepatocyte cell lines could be attributed to biological differences between primary hepatocytes and mitotically active hepatocyte cell lines.

HC-04 cell line

HepG2 cell line

Huh7.5 cell line

Hepa1.6 cell line

6. Lipotoxicity in the liver is widely recognized as a contributor to insulin resistance. The observation that the LMKO liver exhibits reduced lipid accumulation and lipoatrophy compared to control livers suggests a potential improvement in insulin resistance induced by a high-fat diet (HFD) in LMKO mice. However, it is noteworthy that no significant difference in insulin sensitivity was observed between the control and LMKO mice. To gain deeper insights into the insulin sensitivity of liver tissues, it would be beneficial to investigate the insulin signaling

pathway, including the phosphorylation status of key components such as IRS, IR, and AKT. By examining the phosphorylation levels of these proteins, a more comprehensive understanding of the impact of hepatic MTFP1 deletion on insulin signaling and sensitivity in the liver can be obtained.

We thank the reviewer for their suggestion. We did not detect genotype-specific differences in the steady-state levels of proteins belonging to the GOBP_INSULIN_RECEPTOR_SIGNALING_PATHWAY (MM5157) (GSEA) ADIPOR1, AGT, AHSG, AKT1, AKT2, ANKRD26, APC, APPL1, BAIAP2L1, BCAR1, BCAR3, C1QTNF12, C2CD5, CAV2, CCND3, CDK4, CSRP3, CTSD, IGF2, DNAI1, RAF1, EIF4EBP1, EIF4EBP2, ENPP1, ERFE, FER, FFAR3, FOXC2, FOXO1, FOXO4, FUT7, GKAP1, GNAI2, GPLD1, GPR21, GRB10, GRB14, GRB2, GRB7, GSK3A, GSK3B, HRAS, IDE, IGF1R, IL1B, INPP5K, INPPL1, INS1, INS2, INSR, IRS1, IRS2, IRS3, IRS4, KANK1, KL, LEP, LONP1, MAP2K1, MAPK1, MAPK3, MARCKS, MFN2, MIR143, MIR494, MSTN, MZB1, NCK1, NCL, NCOA5, NDEL1, NR1H4, NUCB2, NUCKS1, OBP2A, OGT, OPA1, OSBPL8, PHIP, PDK2, PDK4, PDPK1, PIK3CA, PIK3R1, PIK3R2, PIK3R3, PIP4K2A, PIP4K2B, PIP4K2C, PAK1,, PID1, PRKAA1, PRKCA, PRKCB, PRKCD, PRKCQ, PRKCZ, PTPN1, PTPN11, PTPN2, PTPRA, PTPRE, PTPRF, PTPRJ, RARRES2, RBM4, RELA, RHOQ, RPE65, RPS6KB1, SERPINA12, SESN3, SH2B2, SHC1, SIK2, SIRT1, SLC27A4, SLC2A8, SLC39A14, SMARCC1, SNX5, SOCS1, SOCS3, SOCS7, SOGA1, SORBS1, SORL1, SOS1, SOS2, SRC, SRSF3, STXBP4, TNF, TNS2, SREBF1, TRIM72, TSC2, VWA2, ZBTB7B, ZFP106, ZFP592 in liver biopsies by proteomics. PLCB1 and GSK3A were modestly reduced in LMKO livers. PLCB1 does not play a key role on insulin homeostasis while GSK3A plays a role in glycogen synthase, but it is the only subunit from the pathway deregulated. These results suggest that MTFP1 ablation does not directly impact insulin homeostasis.

Insulin responsive proteins

Our existing bulk RNAseq studies performed on livers of LMKO mice did not observe differences in insulin-responsive gene expression of the following genes (see graph below) (Supplemental Dataset 1), indicating that there is no evidence at the transcriptional level of altered insulin signaling in the liver of LMKO mice. As suggested by the reviewer, we next

examined sought to examine the phosphorylation status of key components of the insulin signaling pathway by western blots using liver lysates from fasted mice stimulated with insulin. We did not observe significant differences in the steady-state levels of AKT and MAPK between control and LMKO livers nor their respective phosphorylation (Fig. S7E, F), indicating that insulin signaling in the liver was not affected by MTFP1 ablation. To validate the specificity of our antibodies, we used C2C12 mouse myoblasts and primary hepatocytes, insulin-stimulated or not, in vitro. Unfortunately, the antibodies directed against phosphorylated IRS and IR failed to reveal a specific signal. Therefore, our data indicate that insulin signaling in the liver of LMKO mice is intact.

7. It would be interesting to explore how the liver-specific MTFP1 KO affects white adipose tissue (WAT) and brown adipose tissue (BAT) to gain a more comprehensive understanding of the systemic effects of hepatic MTFP1 deletion.

We agree with the reviewer, which is why we originally measured WAT tissue mass (Fig. S5E, F) and whole-organism fat mass by NMR minispec (Fig. 5E), neither of which was impacted by MTFP1 ablation under normal chow diet. Furthermore, we did not detect genotype-specific differences in adipocyte area either under normal chow diet or high fat diet (Fig. S7G,H). The treatment of control mice with HFD, which led to increased eWAT mass and adipocyte area, served as a positive control for this experiment. Together these data indicate that WAT is not affected by liver-specific ablation of MTFP1 and we have now added a sentence in the discussion to highlight this observation (line 412).

Regarding brown adipose tissue (BAT), we have now performed new experiments on isolated BAT tissue (Fig. S7J) in which we measured the expression of *Ucp1*, a specific marker of BAT²¹, and *Prdm16*, which is an essential transcription factor required to maintain BAT morphology and thermogenesis in mice²². Under normal housing conditions, we observed no genotype-specific differences in these BAT markers. While we agree that it would be interesting to study thermogenesis in LMKO mice, we do not have the infrastructure in our animal facility to carry out cold challenge tests²³ at 4°C for the moment. Indeed, metabolic crosstalk in tissue-specific knockout mice is possible, however based on an extensive search of the literature, we have not identified liver-specific mitochondrial knockout mouse models that present aberrant BAT-dependent thermogenesis defects.

8. The manuscript does not show a difference in respiratory exchange ratio (RER) between the dark and light cycles, despite the expectation of increased RER during the dark cycle due to more food intake. It would be valuable to discuss this discrepancy and provide potential explanations for the observed results.

Under normal chow diet, both control and LMKO mice do indeed show differences in RER, which we did not originally statistically compare. Our original comparisons were between

genotypes rather than between time points. Now, we have re-analyzed these original data and can show that when comparing the mean RER in light versus dark or specifically at ZT6 versus ZT18, there is a significant difference within but not between genotypes and this is matched by food intake measurements. However, under HFD, the light/dark effect on RER is lost equally in both control and LMKO mice. This HFD-dependent effect has been reported by others Fig. 2G²⁴ and is therefore consistent with our expectations. We have included these new statistical comparisons in the revised version of manuscript in Fig. S1C and 5A.

Thanks to the reviewer's keen eye, we reevaluated the raw data of the cumulative food intake under HFD, which lasted for 5 consecutive days. We noticed that for LMKO mouse number 4, the food intake sensor failed to detect a change beginning at the 3rd day, biasing the measurement of food intake cumulated over 5 days and explaining this outlier. We have now included these raw data to illustrate this error (raw data can be found on columns U,V,W,X,Y,Z,AA and AB of sheet Fig. S6A of the excel file Patitucci et al. Raw_data) and have therefore elected to represent the daily mean food intake during the light and dark periods, rather than the cumulative amounts. This is clearly a technical error of the food sensor as all other concomitant measurements (water intake, distance, speed), whose recording is independent of the mechanics of the food sensor behaved normally over the 5 day recording period. When we calculate daily food intake during the light and dark periods of mice on HFD, we observe no difference between genotypes regardless of whether mean food intake values for the dark and light periods are calculated over 4 days or 2 days. These new graphical representations of both the NCD and HFD mice now appear as Fig. 1J and Fig. S7A-C, respectively. We apologize for this oversight and are grateful that the reviewer helped us detect this error during the revision process.

Minor comments:

1. The manuscript mentions that MTFP1 deletion in primary hepatocytes increases the activity of Complex IV and V, as well as fatty-acid-derived energy metabolism. However, the proteomics data only identified ATP5B and not other proteins such as NDUFA9, SDHA, and MTCO2 (Fig. 2I and J).

To correct this misunderstanding, we wish to highlight that Fig. 2I and J do not refer to the MS-based proteomic evaluation of mitochondrial proteins, but to the quantification of SDS-PAGE and BN-PAGE immunoblots, respectively. In these immunoblots, only a handful of marker proteins for the OXPHOS complexes were employed, such as NDUFA9, SDHA, MTCO2, and ATP5B. ATP5B, but not the other OXPHOS proteins, were shown to be increased in both SDS-PAGE and BN-PAGE which is fully consistent with the global increase in complex V subunits quantified by relative complex abundance measurements from the liver proteomics (Supplemental Dataset 2).

It would be helpful to explain why Complex IV and fatty-acid-derived energy metabolism were increased by MTFP1 deletion without affecting the levels of these proteins.

Indeed, it is intriguing that Complex IV activity can be increased without a commensurate increase in the abundance of this complex (Fig. 2H right, 2J) nor its individual subunits (Fig. 2H, I). Previous pharmacological studies in rodent liver mitochondria have demonstrated that it is possible to acutely and specifically increase Complex IV activity by at least 200% without a commensurate increase in cytochromes a and a3²⁵. Since the Complex IV activity measurements (Fig.2E) were performed in intact isolated liver mitochondria in the presence of Antimycin A, TMPD, and Ascorbate, we can exclude altered electron flux as a determinant of activity. Rather, the most parsimonious explanation for increased Complex IV activity is the remodeling of (inner) mitochondrial membranes by MTFP1 ablation. We did not observe gross alterations in cristae structure in electron micrographs from LMKO livers and BN-PAGE analysis of respiratory chain supercomplexes immunodetected with an anti-COX4 antibody

did not reveal an increase, suggesting that remodeling of Complex IV activity could be linked to a yet unidentified post-translation modification of activity.

2. In Figures 2E and 2F, MTFP1 deletion led to a 200% increase in the activities of Complex IV and V. It would be valuable to explain and discuss how this increase translates to a 200% increase in respiration, as shown in Figure 2B.

The “200% increase in respiration” that the reviewer is referring to concerns the state 3 respiration of LMKO liver mitochondria supplied with palmitoylcarnitine, malate, and ADP. However, state 3 respiration with PGM and SUCC/ROT is far less elevated and commensurate with the Complex IV activity reported in Fig. 2E. This observation is consistent with both a general increase both in Complex IV and V activity (which we show) as well as an increased rate of beta oxidation, namely that of acylcarnitines that could only be detected when exogenously supplied. This point was already discussed in line 131 of the text, which initially read “Interestingly, we observed a 200% increase in state 3 respiration in the presence of palmitoyl carnitine, a fatty acid ester derivative, pointing to an increased efficiency of fatty-acid derived energy metabolism caused by hepatocyte-specific deletion of *Mtfp1*.”

3. While the omics data presented in the manuscript are insightful, there is a need for better integration and linkage of these data to provide a more cohesive and comprehensive understanding of the role of MTFP1 in hepatic mitochondria. It would be beneficial to discuss the relationships between the different omics findings and how they collectively contribute to the overall conclusions of the study.

We have now included a section in the discussion describing the relationship between the OMICS data and how they provide insights into MTFP1 biology in the liver (lines 252-254).

4. In Figure 3C, no bands for MTFP1-binding proteins were observed. It would be advantageous to improve the figure quality to ensure clear visualization of the protein bands and their interactions with MTFP1.

The reviewer is mistaken as the panel labeled “stain-free” is in fact the loading control for the gel, not individual proteins bands of interacting proteins. Interacting proteins were identified by mass spectrometry. The figure quality is at maximal resolution and the original raw data, including this figure, has been submitted to the journal as file “Stain_free_figure3B”.

Reviewer #2 (Remarks to the Author):

*I read with attention the paper by C. Patitucci et al. and I enjoyed the highly instructive content enriched with several explanatory experiments. The paper provides evidence of the importance of mitochondrial function in liver steatosis and the concept that mitochondrial function must be preserved in conditions accumulating intrahepatic fat. In particular, the authors identify Mitochondrial Fission Process 1 protein (MTFP1) as a key regulator of mitochondrial and metabolic activity in the liver. They show that deletion of *Mtfp1* in hepatocytes is physiologically benign in mice but leads to the upregulation of oxidative phosphorylation (OXPHOS) activity and mitochondrial respiration. This effect is, independent of mitochondrial biogenesis. Hepatocyte-specific knockout mice are protected against high fat diet-induced hepatic steatosis and metabolic dysregulation. The deletion of *Mtfp1* in liver mitochondria inhibits mitochondrial permeability transition pore opening in hepatocytes, conferring protection against apoptotic liver damage in vivo and ex vivo. The function of MTFP1 in the liver points to its regulatory function of OXPHOS. The methodology is well described and rather complete, and results explained satisfactorily within each paragraph. The findings are novel, although some translational concepts will increase the impact of this paper. I provide some comments which will help the authors to strengthen their work.*

General

The discussion requires few lines of comments about the potential translational value of the major findings. At the end of the abstract the authors conclude that “...MTRF1.....positioning this gene as an unexpected regulator of OXPHOS and a therapeutic candidate for NAFLD. The same comment needs to be extended at the end of the discussion.

We have now extended the discussion of the potential translational value of the major findings with additional text beginning on line 480 that reads:

“Nevertheless, LMKO mice have availed themselves to be a unique tool to understand how enhancing mitochondrial respiration can combat metabolic liver disease characterized by hepatic steatosis. In humans, 20% to 27% of individuals manifesting steatosis will progress to steatohepatitis and of those, 10% will develop cirrhosis and, eventually hepatocellular carcinoma^{26 27}. Defining the contribution of mitochondrial dysfunction(s) in the progression from simple steatosis to hepatocellular carcinoma has been challenging given the interdependence of the various signaling and metabolic functions of mitochondria. Here, our study provides a tractable genetic target able to enhance respiration and limit free fatty acid-induced hepatic lipid accumulation independently of promoting mitochondrial biogenesis and without uncoupling mitochondria. Hence the mitochondrial mechanism through which MTRF1 ablation protects against hepatic steatosis appears to be functionally distinct from the aforementioned pathways currently being pursued as therapeutic interventions for MASLD and associated pathologies²⁸⁻³⁰. Currently, there is limited information on the beneficial effects in the liver of boosting both Complex IV and V activities concomitantly and further studies are currently underway to determine the breadth of protection that MTRF1 ablation in hepatocytes confers in the context of NASH and may prove useful for the in vivo exploration and development of therapeutic targets for other inherited and acquired liver diseases.”

Abstract

-the text is unbalanced with too much background. Please describe more methods/results.

The abstract length is ultimately limited to 200 words by the journal. Notwithstanding, we have modified the revised the abstract (also considering the new MASLD nomenclature) to better describe the methods and results.

Introduction

- The authors mention the impact of NAFLD but they should also mention the debate about new definitions like MAFLD or even more recently MASLD. This is important in the overall classification of liver steatosis without using “stigmatising” words like “alcoholic” and “fatty”. Please consider these papers doi 10.1097/hep.0000000000000520 PMID 01515467-990000000-00488; doi 10.1007/s11739-023-03203-0 PMID 36807050. After that, please decide what will be the acronym adopted throughout the text, i.e., NAFLD?, MAFLD?, MASLD? Etc.

We thank the reviewer for pointing out these new developments in the field of hepatology. The title of the manuscript references hepatic steatosis rather than NAFLD, MAFLD, and MASLD and thus avoids the use of potentially stigmatizing words like alcoholic and fatty. Notwithstanding, while we do not feel that our manuscript is the appropriate forum to debate human disease nomenclature, we understand that the new nomenclature has been proposed to replace NAFLD by MAFLD, for "metabolic dysfunction-associated fatty liver disease" (MAFLD). This term was initially introduced as a means to emphasize the role of cardiometabolic risk elements in the advancement and progression of liver disease, including cases with other liver disorders³¹, which was further studied in 2023 as the reviewer points out in the two references we had overlooked^{32,33}. As far as we understand, the American Association for the Study of Liver Diseases and the European Association for the Study of

Liver Diseases have recently, formally embraced this terminology in June 2023, after the original submission of our manuscript. We have **therefore replaced NAFLD with MASLD in the text**. We have also **replaced the term NASH with MASH**, for Metabolic dysfunction-associated steatohepatitis.

- Please refer to the percent of subgroups having the chance to progress from simple steatosis to steatohepatitis to cirrhosis and, eventually hepatocellular carcinoma.

This specification has now been added to line 496 in the discussion and refers to the percentage of individuals suffering from steatosis that progress onwards to steatohepatitis, cirrhosis, and HCC.

Results/Methods

- Please clarify the difference between “liver-specific MTFP1 KO” and “hepatocyte-specific MTFP1 KO”. Otherwise use same terminology throughout the text

They are one of the same. To avoid confusion, we have harmonized the text and now refer to liver-specific MTFP1 knockout mice as LMKO mice and have highlighted these changes in the revised text. LMKO mice lack MTFP1-specifically in hepatocytes.

Discussion

- Line 344 parenthesis is missing

This error has been corrected.

Figure 6B

- Drawing breaks on the Y axis will increase the visibility of the lowest bars

We have attempted to add single breaks (two segments) to increase the visibility of the NaCl samples, but this has not fundamentally improved the visibility of the lowest bars. We have therefore elected to maintain the original representation.

Figure 7

The figure is nicely depicted, although I suggest being more detailed about mechanisms and pathways. This would be more in line with the amount of experimental data accumulated in the paper.

We have added additional details to our working model highlighting that increased OXPHOS results from increased activities of complex IV and V. Moreover, we have further illustrated the relationship between MTFP1 and mPTP as well as the inner membrane localization of MTFP1 itself and its function as a negative regulator of Complex IV and V activity.

References:

1. Donnarumma, E. *et al.* Mitochondrial Fission Process 1 controls inner membrane integrity and protects against heart failure. *Nat. Commun.* **13**, 6634 (2022).
2. Martinez-Lopez, N. *et al.* mTORC2-NDRG1-CDC42 axis couples fasting to mitochondrial fission. *Nat. Cell Biol.* **25**, 989–1003 (2023).
3. Jacobi, D. *et al.* Hepatic Bmal1 Regulates Rhythmic Mitochondrial Dynamics and Promotes Metabolic Fitness. *Cell Metab.* **22**, 709–720 (2015).
4. Anand, R. *et al.* The i-AAA protease YME1L and OMA1 cleave OPA1 to balance mitochondrial fusion and fission. *J. Cell Biol.* **204**, 919–929 (2014).
5. Wei, Q. *et al.* MicroRNA-668 represses MTP18 to preserve mitochondrial dynamics in ischemic acute kidney injury. *J. Clin. Invest.* **128**, 5448–5464 (2018).
6. Peralta, S. *et al.* ATAD3 controls mitochondrial cristae structure in mouse muscle, influencing mtDNA replication and cholesterol levels. *J. Cell Sci.* **131**, (2018).

7. Desai, R. *et al.* ATAD3 gene cluster deletions cause cerebellar dysfunction associated with altered mitochondrial DNA and cholesterol metabolism. *Brain* **140**, 1595–1610 (2017).
8. Ishihara, T., Ban-Ishihara, R., Ota, A. & Ishihara, N. Mitochondrial nucleoid trafficking regulated by the inner-membrane AAA-ATPase ATAD3A modulates respiratory complex formation. *Proc. Natl. Acad. Sci. U. S. A.* **119**, e2210730119 (2022).
9. Lepelley, A. *et al.* Enhanced cGAS-STING-dependent interferon signaling associated with mutations in ATAD3A. *J. Exp. Med.* **218**, (2021).
10. Schuettpelz, J., Janer, A., Antonicka, H. & Shoubridge, E. A. The role of the mitochondrial outer membrane protein SLC25A46 in mitochondrial fission and fusion. *Life Sci Alliance* **6**, (2023).
11. Gutiérrez-Aguilar, M. & Baines, C. P. Physiological and pathological roles of mitochondrial SLC25 carriers. *Biochem. J.* **454**, 371–386 (2013).
12. Camacho, J. A. *et al.* Hyperornithinaemia-hyperammonaemia-homocitrullinuria syndrome is caused by mutations in a gene encoding a mitochondrial ornithine transporter. *Nat. Genet.* **22**, 151–158 (1999).
13. Reid, E. S. *et al.* Mutations in SLC25A22: hyperprolinaemia, vacuolated fibroblasts and presentation with developmental delay. *J. Inherit. Metab. Dis.* **40**, 385–394 (2017).
14. Lee, J.-S. *et al.* Loss of SLC25A11 causes suppression of NSCLC and melanoma tumor formation. *EBioMedicine* **40**, 184–197 (2019).
15. Molinari, F. *et al.* Impaired mitochondrial glutamate transport in autosomal recessive neonatal myoclonic epilepsy. *Am. J. Hum. Genet.* **76**, 334–339 (2005).
16. Iacobazzi, V. *et al.* Molecular and functional analysis of SLC25A20 mutations causing carnitine-acylcarnitine translocase deficiency. *Hum. Mutat.* **24**, 312–320 (2004).
17. Zhang, H. *et al.* Elevated mitochondrial SLC25A29 in cancer modulates metabolic status by increasing mitochondria-derived nitric oxide. *Oncogene* **37**, 2545–2558 (2018).
18. Bresciani, N. *et al.* The Slc25a47 locus is a novel determinant of hepatic mitochondrial function implicated in liver fibrosis. *J. Hepatol.* **77**, 1071–1082 (2022).
19. Shi, X. *et al.* Combinatorial GxGxE CRISPR screen identifies SLC25A39 in mitochondrial glutathione transport linking iron homeostasis to OXPHOS. *Nat. Commun.* **13**, 1–15 (2022).
20. Heide, H. *et al.* Complexome profiling identifies TMEM126B as a component of the mitochondrial complex I assembly complex. *Cell Metab.* **16**, 538–549 (2012).
21. Enerbäck, S. *et al.* Mice lacking mitochondrial uncoupling protein are cold-sensitive but not obese. *Nature* **387**, 90–94 (1997).
22. Harms, M. J. *et al.* Prdm16 is required for the maintenance of brown adipocyte identity and function in adult mice. *Cell Metab.* **19**, 593–604 (2014).
23. Lagouge, M. *et al.* Resveratrol improves mitochondrial function and protects against metabolic disease by activating SIRT1 and PGC-1alpha. *Cell* **127**, 1109–1122 (2006).
24. Kim, S. *et al.* Tanycytic TSPO inhibition induces lipophagy to regulate lipid metabolism and improve energy balance. *Autophagy* **16**, 1200–1220 (2020).
25. Desai, S. D., Chetty, K. G. & Pradhan, D. S. Dimethyl sulfoxide elicited increase in cytochrome oxidase activity in rat liver mitochondria in vivo and in vitro. *Chem. Biol. Interact.* **66**, 147–155 (1988).
26. Powell, E. E., Wong, V. W.-S. & Rinella, M. Non-alcoholic fatty liver disease. *Lancet* **397**, 2212–2224 (2021).
27. Friedman, S. L., Neuschwander-Tetri, B. A., Rinella, M. & Sanyal, A. J. Mechanisms of NAFLD development and therapeutic strategies. *Nat. Med.* **24**, 908–922 (2018).
28. Shum, M., Ngo, J., Shirihai, O. S. & Liesa, M. Mitochondrial oxidative function in NAFLD: Friend or foe? *Mol Metab* **50**, 101134 (2021).
29. Pafili, K. & Roden, M. Nonalcoholic fatty liver disease (NAFLD) from pathogenesis to treatment concepts in humans. *Mol Metab* **50**, 101122 (2021).
30. Fromenty, B. & Roden, M. Mitochondrial alterations in fatty liver diseases. *J. Hepatol.* **78**, 415–429 (2023).

31. Eslam, M. *et al.* A new definition for metabolic dysfunction-associated fatty liver disease: An international expert consensus statement. *J. Hepatol.* **73**, 202–209 (2020).
32. Portincasa, P. NAFLD, MAFLD, and beyond: one or several acronyms for better comprehension and patient care. *Intern. Emerg. Med.* **18**, 993–1006 (2023).
33. Rinella, M. E. *et al.* A multi-society Delphi consensus statement on new fatty liver disease nomenclature. *Hepatology* (2023) doi:10.1097/HEP.0000000000000520.

REVIEWERS' COMMENTS

Reviewer #2 (Remarks to the Author):

The manuscript NCOMMS-23-20890A has much improved and I congratulate with the authors for the detailed answers provided to both reviewers.